

# Regime-oriented causal model evaluation of Atlantic-Pacific teleconnections in CMIP6

Soufiane Karmouche[1,2], Evgenia Galytska[1,2], Jakob Runge[3,4], Gerald A. Meehl[5], Adam S. Phillips[5], Katja Weigel[1,2], and Veronika Eyring[2,1]

[1]University of Bremen, Institute of Environmental Physics (IUP), Bremen, Germany
[2]Deutsches Zentrum für Luft- und Raumfahrt (DLR), Institut für Physik der Atmosphäre, Oberpfaffenhofen, Germany
[3]Deutsches Zentrum für Luft- und Raumfahrt (DLR), Institut für Datenwissenschaften, Jena, Germany
[4]Technische Universität Berlin, Berlin, Germany
[5]National Center for Atmospheric Research (NCAR), Boulder, CO, USA

**Correspondence:** Soufiane Karmouche (sou_kar@uni-bremen.de)

**Abstract.**

   The climate system and its spatio-temporal changes are strongly affected by modes of long-term internal variability, like the Pacific Decadal Varibility (PDV) and the Atlantic Multidecadal Variability (AMV). As they alternate between warm and cold phases, the interplay between PDV and AMV varies over decadal to multidecadal timescales. Here, we use a causal discovery method to derive fingerprints in the Atlantic-Pacific interactions and investigate their phase-dependent changes. Dependent
on the phases of PDV and AMV, different regimes with characteristic causal fingerprints are identified in reanalyses in a first step. In a second step, a regime-oriented causal model evaluation is performed to evaluate the ability of models participating in the Coupled Model Intercomparison Project Phase 6 (CMIP6) in representing the observed changing interactions between PDV, AMV and their extra-tropical teleconnections. The causal graphs obtained from reanalyses detect a direct opposite-sign response from AMV on PDV when analysing the complete 1900-2014 period, and during several defined regimes within that
period, for example, when AMV is going through its negative (cold) phase. Reanalyses also demonstrate a same-sign response from PDV on AMV during the cold phase of PDV. Historical CMIP6 simulations exhibit varying skill in simulating the observed causal patterns. Generally, Large Ensemble (LE) simulations showed better network similarity when PDV and AMV are out of phase compared to other regimes. Also, the two largest ensembles (in terms of number of members) were found
to contain realizations with similar causal fingerprints to observations. For most regimes, these same models showed higher network similarity when compared to each other. This work shows how causal discovery on LEs complements the available diagnostics and statistics metrics of climate variability to provide a powerful tool for climate model evaluation.

*Copyright statement.* TEXT





## 1 Introduction

Modes of natural climate variability from interannual to multidecadal timescales have large effects on regional and global climate with important socio-economic impacts. Despite their importance, systematic evaluation of climate models and their simulation of internal variability remains a challenging task (Eyring et al., 2019). The available observational datasets are not only short in time, but also hold considerable uncertainties that arise from errors in the data record during the pre-satellite era (Phillips et al., 2014; Fasullo et al., 2020; Eyring et al., 2021). Generally, in order to test their performance, the models are often

compared to reanalysis datasets based on observations. This approach is a key to estimate the ability of models to correctly simulate internal variability. An evaluation study by Fasullo et al. (2020) showed a systematic improvement in the representation of modes of climate variability through the different phases of the Coupled Model Intercomparison Project (CMIP), where models largely capture the statistical properties of these modes (e.g., timescale, autocorrelation, spectral characteristics, and spatial patterns). However, across the CMIP archive, comparisons with observations also reveal remarking systematic errors.

These are errors that have only little or no improvement due to the complexity of the climate system and the difficulty to assign a specific cause to a specific systematic error or bias (Stouffer et al., 2017; Fasullo et al., 2020; Eyring et al., 2021).

It is therefore a priority to go beyond spatial and spectral properties and apply new approaches that reveal whether a climate model correctly simulates the observed lagged teleconnections between remote regions. Here, causal discovery methods provide a way to estimate such dynamical climate dependencies from data timeseries (Ebert-Uphoff and Deng, 2012; Runge et al.,

2019b; Runge, 2020; Runge et al., 2019a; Nowack et al., 2020). Causal graphs not only help to assess the degree to which a climate model recreates well-defined connections within the climate system, but also to determine if specific phenomena are simulated for the right reasons. As the nature of these connections and phenomena is supposed to vary depending on the state of multidecadal processes of internal climate variability, we investigate the causal relations not only for the complete historical period, but also for shorter, state-dependent timescales that define different regimes of dependencies.

In this study, we utilize a regime-oriented causal analysis on indices of dominant modes of long-term variability over the Atlantic and Pacific to investigate the interactions between the two basins in CMIP Phase 6 historical simulations (CMIP6, Eyring et al., 2016) as well as in Large Ensembles (LE) and compare those results to reanalysis data. To do so, we first calculate the two leading modes of multidecadal coupled (ocean-atmosphere) climate variability over the Pacific and Atlantic: Pacific Decadal Variability (PDV) and Atlantic Multidecadal Variability (AMV). PDV, encompassing a symmetric variability

pattern over the North and South Pacific (Mantua et al., 1997; Chen and Wallace, 2015), with an El Niño-Southern Oscillation-(ENSO) like decadal variability over the tropical Pacific extending over the entire Pacific basin (Nitta and Yamada, 1989; Zhang et al., 1997; Meehl et al., 2013), can be defined by Pacific sea surface temperature (SST) anomaly fields. Its influence on the other hand, expands well beyond the Pacific affecting regional- and global-scale climate on decadal timescales. Its temporal evolution is characterized by an interannual and decadal variability with some pronounced shifts, notably the extensively

studied 1976/77 transition (Zhang et al., 1997; Power et al., 1999; Mantua et al., 1997; Arblaster et al., 2002; Meehl et al., 2009). In particular, Ebbesmeyer et al. (1991) identified dramatic changes in the North Pacific biota and climatic variables during that period. The positive PDV phase dominated during the period from the mid-1970s through late 1990s, while the





following period of global warming hiatus entailed a switch to negative phase (Meehl et al., 2016; Fyfe et al., 2016). The second dominant pattern of internal multidecadal variability, the AMV, acts on the North Atlantic region. Sometimes referred

to as the Atlantic Multidecadal Oscillation (AMO, Kerr 2000), AMV is characterized by a dipole SST variability pattern featuring opposite sign anomalies between the Tropical North Atlantic and South Atlantic (IPCC, 2021). Index timeseries of the observed AMV pattern show that the mode goes through preferred phases for multidecadal periods with the positive phase persisting since the late 1990s to nowadays. The AMV was also discovered to have significant socio-economic and climate impacts, particularly on the Indian summer monsoon, North American and European summer climate and hurricanes over the

Atlantic (Folland et al., 1986; Sutton and Hodson, 2005; Knight et al., 2006; Zhang and Delworth, 2006; Si and Hu, 2017; Yan et al., 2017).

Previous research focused on Atlantic-Pacific interactions suggests changing forcing mechanisms can be applied by one basin on the other (d'Orgeville and Peltier, 2007; Wu et al., 2011; Kucharski et al., 2016; Nigam et al., 2020). Observational analyses concluded that the multidecadal component of the negative PDV phase can lag the positive AMV phase by about a

decade (Zhang and Delworth, 2007; d'Orgeville and Peltier, 2007). Literature suggests a PDV-AMV link through a tropical pathway where increasing Atlantic temperatures instigate a La Niña-like cooling in the equatorial Pacific, and a consequent weakened Aleutian low in the North (McGregor et al., 2014; Kucharski et al., 2016; Li et al., 2016; Ruprich-Robert et al., 2017). Meehl et al. (2021a) showed that the Atlantic and Pacific are mutually and sequentially interactive and are connected mainly through the atmospheric Walker circulation with some extra-tropical contributions. Components of the PDV in that

study were found to be linked to Aleutian low variability associated with the Pacific-North American (PNA) pattern (Wallace and Gutzler, 1981), a prominent mode over the Northern Hemisphere extra-tropics, with a quadrupole anomaly field of 500 hPa geopotential height (H500) that can influence the subtropical North Atlantic. Teleconnections to the Southern Hemisphere were also noted by Meehl et al. (2021a) involving the Pacific-South-American (PSA) pattern that ends up influencing the subtropical South Atlantic. Another study involving coupled model simulations from Zhang et al. (2018) agreed with Meehl et al. (2021a)

and showed that the PSA, which can be thought of as the South Pacific counterpart of PNA, generates a forcing that translates into the Southern Hemisphere component of PDV. To assess these possible extra-tropical connections, in addition to PDV and AMV, we include in our causal discovery study indices for both PNA and PSA modes. The indices of the latter modes are both based on sea level pressure (SLP) anomalies. PSA is generally expressed through two modes, in this study we use PSA mode 1 (PSA1) index as the second Empirical Orthogonal Function (EOF) of area-weighted SLP anomalies in the South Pacific (Mo

and Higgins, 1998, see Methods).

Figure 1 shows the various steps of our regime-oriented causal model evaluation approach presented in this paper, which we organise as follows: Sect. 2 describes methods (Sect. 2.1) and data (Sect. 2.2) that were used in this study. In Sect. 2.1.1 we present the package used to generate the indices and spatial patterns of the different modes of climate variability. This is followed by an introduction to the causal discovery method (Sect. 2.1.2), and the framework for the regime-oriented causal

model evaluation (Sect. 2.1.3). In Sect. 2.1.4 we introduce the causal network comparison method via calculation of $F_1$-scores. The analysed reanalysis datasets and CMIP6 models used in this study are listed in Sect. 2.2. The Results (Sect. 3) start with a correlation analysis to compare the SST and SLP regression maps associated with the CMIP6-simulated timeseries of AMV,





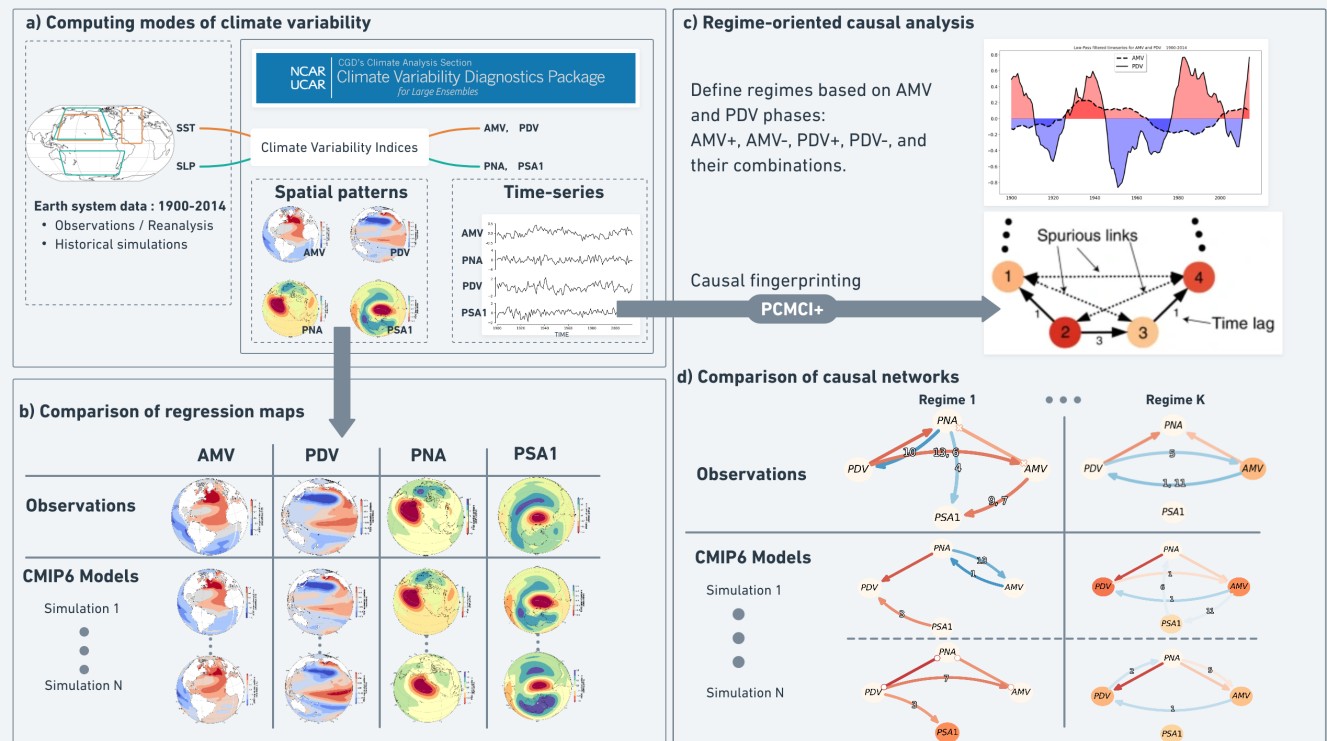

**Figure 1.** Framework for the regime-oriented causal model evaluation. **(a)** Gridded SST and SLP data used to calculate indices for AMV, PDV, PNA and PSA1 modes of climate variability. Diagnostics from the NCAR Climate Variability Diagnostic Package for Large-Ensembles (CVDP-LE) produce the timeseries of these indices and their associated spatial patterns (regression maps). **(b)** We first, as a sanity check, compare the CMIP6 model-simulated SST (for AMV and PDV) and SLP (for PNA, PSA1) regression maps to those from reanalysis before **(c)** using the timeseries of the four indices for the regime-oriented causal analysis. Here we define different regimes depending on the sign of the 13-year low-pass filtered AMV and 11-year low-pass filtered PDV timeseries. For every regime we run PCMCI+ to estimate instantaneous and lagged links between nodes representing the timeseries of the indices calculated in (a) from the reanalyses and model data. In this schematic example, there are four indices, with node color indicating auto-correlation, and there is a causal link (solid black arrow) between index 2 and indices 1 and 3, and then there is a causal link between indices 3 and 4. The method identifies and removes spurious links (see black dashed arrows) between indices 1 and 4, or 2 and 4. Unitless representative time lags are labeled on each causal link, where index 1 lags index 2 by one time-step (depending on temporal resolution of the timeseries, here yearly), index 3 lags index 2 by three, and index 4 lags index 3 by one. Applying the method to the timeseries in (a) provides **(d)** dataset- and regime-specific causal fingerprints in a similar format to the schematic in (c), which can be used for model evaluation and intercomparison. We calculate annual averages from the monthly timeseries of PDV and AMV provided by CVDP-LE. This way, the dataframe is fit for multi-year and decadal causal estimations. In addition to the subtraction of global mean temperatures in the CVDP-LE calculation of PDV and AMV, the causal networks are estimated after linearly detrending the timeseries of the four indices to ensure their stationarity. The estimated causal dependencies (links) are hence assumed to be a mixture of internal variability and time-varying anthropogenic forcing (mainly from aerosols).



PDV, PNA and PSA to those from reanalysis data (Sect. 3.1). As the causal analysis only uses timeseries information of the calculated indices, this comes as a sanity check to measure the similarity between the observed and simulated spatial patterns
associated with the index timeseries. This is followed by Sect. 3.2 where we show the causal networks from reanalysis data during different regimes. These serve as reference for the regime-oriented causal model evaluation in the subsequent Sect. 3.3. We discuss the results in Sect. 3.4 before closing the paper with a summary in Sect. 4.

## 2   Methods and Data

### 2.1   Methods

#### 2.1.1   Climate Variability Diagnostic Package

Developed by the National Center for Atmospheric Research (NCAR), the Climate Variability Diagnostic Package for Large Ensembles (CVDP-LE) provides an analysis tool for the evaluation of the major modes of internal climate variability tailored for large-ensemble climate models (Phillips et al., 2020). It includes diagnostics to compute indices for the major modes of coupled and large-scale atmospheric climate variability. The package also offers comparison metrics for the spatial and
temporal patterns with respect to reference observational datasets.

For the indices of our selected modes of climate variability (see enumeration below), we use the diagnostic results computed from the CMIP6 LE historical simulations and from reanalyses data over the 1900-2014 period. These are calculated by the CVDP-LE and publicly available as Network Common Data Format (NetCDF) files on the Community Earth System Model (CESM) Climate Variability and Change Working Group's (CVCWG) CVDP-LE Data Repository under
https://www.cesm.ucar.edu/working_groups/CVC/cvdp-le/data-repository.html. We use index timeseries and their associated spatial patterns (SST regression maps for AMV and PDV, PSL regression maps for PNA and PSA1). The indices used in this analysis are computed by the CVDP-LE package as follows:

1. PDV Index (sometimes referred to as the PDO index): is defined as the standardized principal component (PC) timeseries associated with the leading EOF of area-weighted monthly SST anomalies over the North Pacific region [20-70N, 110E-
100W] minus the global mean [70N-60S] (effectively detrending the data). (Mantua et al., 1997)

2. AMV Index (sometimes referred to as the AMO index): is defined as monthly SST anomalies averaged over the North Atlantic region [0-60N, 80W-0W] minus the global mean [60N-60S] (effectively detrending the data). (Trenberth and Shea, 2006)

3. Pacific-North American Pattern (PNA): the leading EOF of area-weighted sea level pressure (SLP) anomalies over the
region [20-85N, 120E-120W]. We use timeseries constructed from yearly winter December-January-February (DJF) means.

4. Pacific–South American Pattern Mode 1 (PSA1) second EOF of area-weighted SLP anomalies south of 20S (Mo and Higgins, 1998). We use timeseries calculated from annual means (ANN).



The idea behind subtracting the global mean in the definition of the SST-based modes, PDV and AMV, is to reduce po-
tential effects of external greenhouse gas (GHG) forcing. The space- and time-varying aerosol forcing, however, is expected
to contribute to the Atlantic and Pacific SST variability represented by the calculated AMV and PDV indices (Booth et al.,
2012; Smith et al., 2016; Watanabe and Tatebe, 2019; Meehl et al., 2021a). According to their CVDP-LE definitions above,
the calculations of PNA and PSA1 do not include any detrending. This is because, in models, the externally-forced component
of these SLP-based modes (unlike the SST-based ones) can generally be neglected when compared to the internally-generated
component (Deser et al., 2012; Phillips et al., 2020). Thus, we presume that the aforementioned indices calculated by CVDP-
LE, although not exhibiting a trend, are a combination of internal variability and external aerosol forcing. This is true for
model-simulated indices and the ones calculated from reanalysis data (with the exception of the observed PSA1 timeseries
which include a noticeable trend, not shown). To compare the simulated spatial patterns to the observed ones (Sect. 3.1), we
correlate the regression maps associated with the timeseries of the indices as they are calculated by the CVDP-LE (see enu-
meration above). Only prior to applying the causal discovery algorithm (Sect. 2.1.2) do we further remove any linear trend that
might still be present in the data to ensure stationarity. Moreover, as the focus of the paper revolves around causal pathways
on decadal (multi-year) timescales, we perform annual averages of the AMV and PDV timeseries as they are computed based
on monthly means by the CVDP-LE. Hence, for all results to be presented in this paper, we maintain the presumption that the
calculated climate variability indices (eventually their spatial patterns and causal fingerprints) represent a mixed response of
internally-generated variability and external aerosol forcing.

### 2.1.2   PCMCI+ Algorithm

For the regime-oriented causal analysis, we use a Python package called Tigramite, freely available at https://github.com/
jakobrunge/tigramite, designed to efficiently estimate causal graphs from timeseries datasets. The causal discovery framework
within Tigramite is called PCMCI (Peter Clark Momentary Conditional Independence) (Runge et al., 2019b). Its suitability for
the challenges of timeseries data as studied here, mainly high dimensionality due to the number of variables and time lags, as
well as autocorrelation, was studied in Runge et al. (2019b); Runge (2020). While the PCMCI framework is also suitable for
nonlinear dependencies, in this paper we focus on linear relationships and use an extended version of PCMCI called PCMCI+
that can not only detect lagged (time lag $\tau > 0$), but also contemporaneous ($\tau = 0$) causal links (Runge, 2020).

PCMCI+ consists of two principal phases: a skeleton discovery phase and an orientation phase. Considering a time-dependent
system $(X_t)$ of $N$ variables $\mathbf{X_t} = (X_t^1, ..., X_t^N)$, the skeleton discovery starts first by applying the PC1 Markov set discovery
algorithm which is based on the PC algorithm (named after its inventors, Peter Spirtes and Clark Glymour) on a completely
connected graph. The iterative PC1 algorithm tests for every lagged pair of nodes (variables) $(X_{t-\tau}^i, X_t^j)$ whether they are
conditionally independent on efficiently selected conditions of other lagged variables, and, if so, removes the adjacency be-
tween them. The lagged conditions at this stage serve to estimate for each variable $X_t^j$ a superset of lagged parents $\widehat{\mathcal{B}}_t^-(X_t^j)$
for which the adjacencies are oriented by time-order. In this step there still can be spurious links due to contemporaneous con-
founders. Hence, in the second skeleton discovery step contemporaneous conditions are iterated over in momentary conditional




independence (MCI) tests implemented with partial correlation:

$$X_{t-\tau}^i \perp\!\!\!\perp X_t^j | \mathcal{S}, \widehat{\mathcal{B}}_t^-(X_t^j) \setminus \{X_{t-\tau}^i\}, \widehat{\mathcal{B}}_{t-\tau}^-(X_{t-\tau}^i)$$

where $\widehat{\mathcal{B}}_t^-(X_t^j)$ are the lagged conditions of $X_t^j$ and $\widehat{\mathcal{B}}_{t-\tau}^-(X_{t-\tau}^i)$ are the (time-shifted) lagged conditions of $X_{t-\tau}^i$ obtained

in the first step. By iterating through subsets $\mathcal{S} \subset \mathbf{X_t}$ of contemporaneous adjacencies, the algorithm fully removes spurious links. The partial correlation tests assume a $t$-statistic with degrees of freedom given by the effective sample size $n-2-|\mathcal{S}, \widehat{\mathcal{B}}_t^-(X_t^j) \setminus \{X_{t-\tau}^i\}, \widehat{\mathcal{B}}_{t-\tau}^-(X_{t-\tau}^i)|$. The result is a graph with lagged and contemporaneous adjacencies. Lagged adjacencies are oriented by time-order since causation can only go forward in time. This skeleton phase is followed by a collider orientation phase, which further orients contemporaneous links based on unshielded triples $X_{t-\tau}^i - X_t^k - X_t^j$ where $\tau \geq 0$. If $X_t^k$ is not

part of the separating set $\mathcal{S}$ that makes $X_{t-\tau}^i$ and $X_t^j$ independent, then the triple is oriented as $X_{t-\tau}^i \to X_t^k \leftarrow X_t^j$. Further contemporaneous links are then oriented such that the graph does not include cycles (see rules R1-R3 in Runge, 2020). The resulting graph then contains directed lagged and contemporanous links, but also unoriented adjacencies indicating that the collider and orientation rules could not be applied (Markov equivalence), or a conflicting adjacency where different rules are conflicting, for example, due to finite sample issues. For visualization purposes the estimated timeseries graph is then

aggregated in a process graph (Figure 2) that summarizes the causal dependencies and their time lags. The link strength can be estimated in different ways, for example as standardized (causal) regression coefficients (Runge et al., 2015; Runge, 2021), but here we use the MCI partial correlation values corresponding to the conditional independence test statistic above.

A full method description of the original PCMCI and its PCMCI+ extension along with their respective pseudo code, proofs of their asymptotic consistency, and numerical experiments can be found in Runge et al. (2019b) and Runge (2020), respec-

tively. These works also explain the underlying assumptions under which the detected links can be interpreted causally. Most importantly, since unobserved confounders can still render links as spurious, the graphs are causal only with respect to the analysed variables. Applying more advanced methods (Gerhardus and Runge, 2020) that can deal with hidden variables would considerably deteriorate the reliability of causal graph inferences for the short sample sizes available here.

Figure 2 demonstrates the application of PCMCI+ algorithm on CVDP-LE datasets. Since causal discovery requires station-

ary timeseries (Runge, 2018), first we consider into our analysis detrended yearly 1900-2014 timeseries of modes of climate variability, namely AMV, PNA, PDV, and PSA1 (left). The resulting causal network from the application of PCMCI+ algorithm is shown in right (Figure 2). The direction, sign, strength (|cross-MCI| value) and time lag ($\tau$) of the estimated causal links are all attributes that can be conveniently read off the generated causal graphs. Each node on a causal network represents a variable and the node color its auto-correlation (self-links of each variable). The link color shows the cross-MCI value which denotes

the sign and strength of the estimated causal link between two variables. The time lag for lagged links (curved arrows) are shown as small labels on the links. For those connections that occur at multiple lags, the color of the link shows the strongest link, but the label depicts all significant lags sorted by their strength. The contemporaneous links are shown as straight arrows ("→"; when directionality is decided), straight lines with circle-shaped ends ("o—o"; when the adjacency indicates a Markov equivalence), or straight lines with cross-shaped ends ("x—x"; indicating conflicting orientation rules).





## Causal network estimation from timeseries information

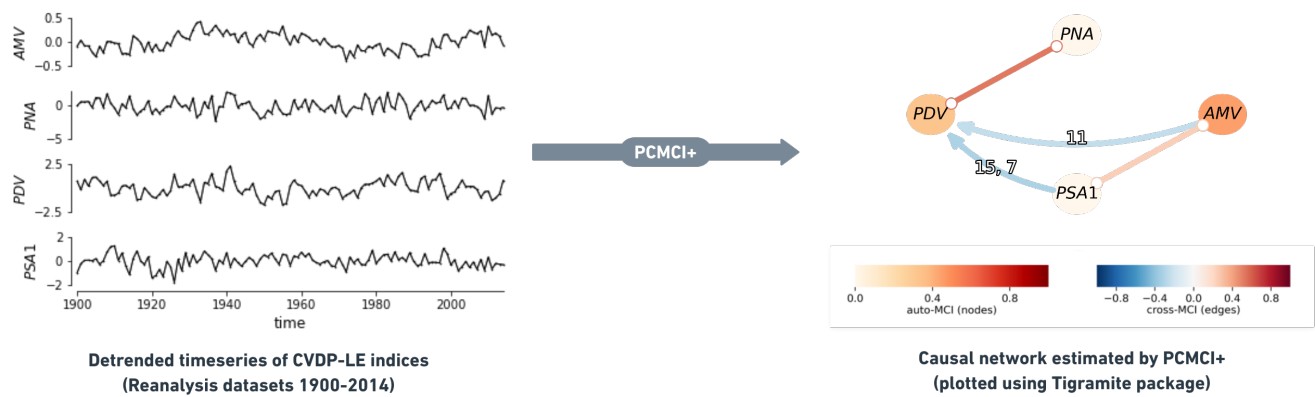

**Figure 2.** Constructing causal network using Tigramite by applying PCMCI+ on timeseries calculated by CVDP-LE from reanalysis datasets. Each node on a causal network (right) represents a variable (timeseries, left) and the node color its auto-correlation (self links of each variable). The link color shows the cross-MCI partial correlation value which denotes the sign and strength of the estimated causal link between two variables. The time lag for lagged links (curved arrows) are shown as small labels on the links. Straight lines represent instantaneous causal links happening with no time lag.

With regards to the parameter settings for the PCMCI+ algorithm, we set the maximum time lag ($\tau_{max}$) to 15 years ( $\tau_{max} = 15$ time-steps, as we are using one data point per year). The significance level of the MCI partial correlation tests above $\alpha_{pc}$ is set to 0.05.

### 2.1.3    Set up for regime-oriented analysis

The teleconnections between the Pacific and Atlantic ocean basins are suggested to follow different regimes depending on

the decadal phases that the AMV and PDV go through (Meehl et al., 2021a). In order to clearly identify the time periods of each phase, we smooth the timeseries data by applying 11-year and 13-year low-pass filters on PDV and AMV, respectively. Figure 3a shows the observed detrended low-pass filtered AMV and PDV timeseries used to specify the different phases and regimes for the masking before applying the PCMCI+ algorithm (the labeled regimes on the timeseries are only three out of the 10 we run the analysis over). First, running the analysis on the complete time period is intended to reveal the consistent causal

dependencies throughout the complete historical timeseries (see Figure 2). The resulting causal networks from the complete period do not, however, expose much information on the causal effects which are changing over shorter time periods depending on how the PDV and AMV are varying during those phases. In order to identify these phase-dependent causal dependencies, we perform the analysis on multiple shorter periods (regimes) by selecting the time-steps that represent either the positive (warm) or negative (cold) phases based on the low-pass filtered indices, with AMV+(-) for when the value of low-pass filtered

AMV is positive (negative), and the same for PDV+(-). We further split these regimes into combinations of warm and cold





PDV and AMV phases (PDV+/AMV+, PDV+/AMV-, PDV-/AMV+, PDV-/AMV-). Additionally, since some regimes are too short to reveal any dependencies, we also opted to run the analysis for an 'In-Phase' regime that sums the PDV+/AMV+ and PDV-/AMV- periods. The remaining time-steps would then consist of the 'Out-of-Phase' regime for the period where the two low-pass filtered indices have opposite signs (PDV+/AMV- and PDV-/AMV+). This means that in addition to running it on the complete period, we apply the PCMCI+ algorithm on 10 different shorter time periods (within the original 1900-2014 period) for each dataset (see Figure 3a for reanalysis data). Figure 3b shows how we use the regimes defined in Figure 3a to mask the timeseries before applying the PCMCI+ method. This is shown for PDV+ and PDV- regimes as example. For each case, the gray shaded parts of the timeseries are masked periods, i.e. only the black shaded periods (see timeseries in Figure 3b) are considered.

We note that the low-pass filtered indices are used only to extract the time periods that constitute each regime. We remove any linear trend that might be present in the data prior to applying the causal discovery algorithm. In this way, the effects of external forcings are reduced. The four indices (AMV, PNA, PDV, PSA1) to which PCMCI+ is applied are represented by detrended yearly unfiltered (not smoothed) timeseries (see Figure 2 and Figure 3b).

### 2.1.4 $F_1$-scores for causal network comparison

To quantify the similarity between the resulting causal graphs (networks) from model simulations and those from observations, we follow a similar modified $F_1$-score as in the methods by Nowack et al. (2020). The $F_1$-score ranges between 0 (no match) and 1 (perfect network match) and is based purely on the existence or non-existence of links in a network relative to a reference network. The $F_1$-score combines the statistical precision (P, fraction of links in model simulation network that also occur in the reference network) and recall (R, fraction of links in the reference network that are detected in the model simulation network) and is defined as:

$$F_1 = \frac{2 \times P \times R}{P + R}$$

with

$$P = \frac{TP}{TP + FP}$$

and

$$R = \frac{TP}{TP + FN}$$

where FP is the number of falsely detected links and FN is the number of not detected links. We modify the definition as in Nowack et al. (2020) so that a link is considered a true positive (TP) if it is found with the same sign of MCI partial correlation as in the reference network. We further relax the time lag constraint by considering a TP to exist if a link is found in a $\pm10$ time-step interval compared to the lag in the reference network (i.e. $[min(\tau_{max}, \tau + 10), max(0, \tau - 10)]$).





**Figure 3. (a)** PDV and AMV timeseries calculated by CVDP-LE diagnostics on ERSSTv5 data are smoothed using 11 and 13-year low-pass filters, respectively. 10 regimes are defined (see table on the left) in addition to the 1900-2014 complete period. The PCMCI+ algorithm is applied on unfiltered (non-smoothed) PDV, AMV, PNA, PSA1 yearly detrended timeseries that are masked according to the periods that define each regime. The right arrows on the smoothed timeseries represent unmasked periods from three out of 10 regimes (PDV+/AMV+, PDV-/AMV+, and PDV+/AMV-). **(b)** The regimes identified in (a) are used to mask the non-smoothed (but detrended) index timeseries before applying PCMCI+. Here, for example, we show how we mask the data according to the PDV- (top) and PDV+ (bottom) regimes. The grey shaded periods are masked and thus not considered during the PCMCI+ analysis. Note that the masking here refers to variables at time point $X_t^j$ while their lagged parents can originate also from a masked period (gray shaded). This setting is referred to as `mask_type='y'` in Tigramite. Consequently, applying PCMCI+ on differently-masked timeseries produces different causal networks (network in top vs network in bottom)





## 2.2 Data

From the 1900-2014 historical climate variability diagnostic results provided by the CVDP-LE, we choose the SST from the Extended Reconstructed Sea Surface Temperature (ERSST) Version 5 (Huang et al., 2017) by the National Oceanic and Atmospheric Administration (NOAA) as reference for the AMV and PDV indices and spatial patterns. Whereas for the PNA and PSA1 modes, we use as reference, SLP from the $20^{th}$ Century Atmospheric Reanalysis extended with ERA5 (ERA20C_ERA5), provided by the European Centre for Medium-Range Weather Forecasts (ECMWF) and assimilating observations of surface pressure. The reference data serve for comparison to evaluate how indices generated using a selection of 12 Large Ensemble CMIP6 historical models reproduce the observed spatial patterns and causal dependencies. The list of CMIP6 LE models (with the number of realization per model) is provided in Table 1.

**Table 1.** CMIP6 Large Ensemble historical simulations used in the analysis

| Dataset | | Components | | N° realisations used | References |
|---|---|---|---|---|---|
| CMIP6 LE | Institute | Atmosphere model | Ocean model | | |
| ACCESS-ESM1-5 | CSIRO | HadGAM2 | ACCESS-OM2 | 10 | Ziehn et al. (2019) |
| CESM2 | NCAR | CAM6 | POP2 | 11 | Danabasoglu (2019) |
| CNRM-ESM2-1 | CNRM | Arpege 6.3 | NEMO3.6 | 10 | Seferian (2018) |
| CanESM5 | CCCma | CanAM5 | NEMO3.4.1 | 65 | Swart et al. (2019) |
| EC-Earth3 | EC-Earth | IFS cy36r4 | NEMO3.6 | 20 | Döscher et al. (2022) |
| GISS-E2-1-H | NASA | GISS-E2.1 | HYCOM Ocean | 23 | Kelley et al. (2020) |
| INM-CM5-0 | INM | INM-AM5-0 | INM-OM5 | 10 | Volodin et al. (2019) |
| IPSL-CM6A-LR | IPSL | LMDZ | NEMO-OPA | 32 | Boucher et al. (2018) |
| MIROC6 | JAMSTEC, AORI, NIES,R-CCS | CCSR AGCM | COCO4.9 | 50 | Shiogama et al. (2019) |
| MPI-ESM1-2-LR | MPI-M | ECHAM6.3 | MPIOM1.63 | 10 | Wieners et al. (2019) |
| NorCPM1 | NorESM Climate modeling Consortium | CAM-OSLO4.1 | MICOM1.1 | 30 | Bethke et al. (2019) |
| UKESM1-0-LL | Met Office Hadley Centre | MetUM-HadGEM3-GA7.1 | NEMO-HadGEM3-GO6.0 | 18 | Tang et al. (2019) |

We note that in the spatial correlation analysis in the next section, monthly averages are used for AMV and PDV as that is the time resolution originally provided by the CVDP-LE for these modes. The diagnostic package does not produce monthly fields for the PNA and PSA1, so we use winter means (DJF) and all-year annual means (ANN), respectively. We found that most model simulations show weak correlations with reanalysis data for the annually averaged PNA (ANN, not shown) compared to the winter averaged PNA (DJF, Table 2). Hence, we chose winter means instead of annual means for PNA to reduce any seasonal bias within the simulated spatial patterns. The spatial patterns do not depend much on the time resolution (yearly or monthly) of the data, as they are all calculated on the whole 1900-2014 period. Prior to applying the causal discovery algorithm (Sect. 3.2), however, we yearly average the AMV and PDV timeseries (computed based on monthly means by the CVDP-LE). This way we unify the time resolution of our data to fit the causal analysis by using the yearly resolution to investigate connections on long timescales.



## 3 Results

### 3.1 Similarities between the simulated and observed spatial patterns

To accompany the causal analysis, we first calculate pattern correlations ($r$) for each simulation's SST and SLP regression maps with respect to the reanalysis regression maps (for the complete 1900-2014 period, see Figure 4a). This is to quantify the similarity between the observed and simulated spatial patterns for the four modes and build credibility that the CMIP6 simulated indices we use in the regime-oriented model evaluation have spatial expressions that resemble those of indices calculated from reanalysis datasets. To introduce a benchmark of model performance, we calculate a Mean Score for each single simulation by taking the average of the four $r$ values (after applying a Fisher z-transform).

To look closer at how the spatial correlation values spread across every LE and how they differ from one climate variability mode to another, Figure 4b provides a color-coded box-plot showing the distribution of these spatial correlation values, and their respective averages across every Large Ensemble of CMIP6 simulations used in the analysis. It depicts the similarity between the observed (reference maps in Figure 4a) and the simulated patterns from the regression maps for the four modes, with values approaching 1 indicating a better simulation of the patterns associated with the observed modes.

Sorted by the ensemble average mean score of every CMIP6 LE, Table 2 provides a view of the distribution (in the form of minimum, mean, maximum) of spatial correlation values for every mode and their Mean Score for every CMIP6 LE model. It can be seen from Figure 4b and Table 2, based on the ensemble average mean score, that most models perform quite well in simulating the observed geographical patterns of the four indices in Figure 4a, with pattern correlations mostly above 0.75. The UKESM1-0-LL (0.80), MIROC6 (0.80), MPI-ESM1-2-LR (0.79), ACCESS-ESM1-5 (0.77) and CanESM5 (0.77) outperform the other CMIP6 LEs in terms of recreating the spatial patterns of the four selected modes of climate variability. The number of ensemble members within every LE has no apparent effect on the spread of the $r$ value distribution across the models. For example, UKESM1-0-LL and MIROC6 with 18 and 50 realizations respectively, share similar narrow interquartile ranges (IQR, the width between the 3rd and 1st quartiles) of $r$ values for the four climate variability spatial patterns. Appendix Table A1 shows the distribution of Pearson $r$ correlation between observed and simulated spatial patterns of PNA, PSA1, PDV, and AMV from a 10th, 50th, 90th percentile perspective. Looking only at the mean score spread, Table A1 shows the 10th-90th percentile value range is 0.78-0.83 for UKESM1-0-LL, and 0.77-0.82 for MIROC6. This means that most members of these two model ensembles agree between each other and show high spatial similarity with observations when simulating the four modes. It can be concluded that the models generally do a good job in simulating the geographical patterns of the different modes but with different precision. Although the models with high mean scores tend to display high pattern correlations with observations for the four modes of climate variability, the white scatter points on Figure 4b imply that they simulate the PNA (purple) atmospheric mode slightly better than its South Pacific counterpart, the PSA1 (cyan) when compared to the ERA20C_ERA5 reference patterns. These high scoring models, notably UKESM1-0-LL, MPI-ESM1-2-LR, MIROC6, CanESM5 and IPSL-CM6A-LR also, on average, simulate better PDV (red) monthly spatial patterns compared to AMV (green), with ERSSTv5 as a reference dataset for the 1900-2014 period. The mean scores of CESM2, GISS-E2-1-H and NorCPM1 are strongly affected by the low correlation coefficients obtained for the PSA1 mode (cyan boxes). The 50th percentile bar on the cyan box for





## a) Reference regression maps

## b) Comparison summary

**Figure 4. (a)** Reference for comparison: SST regression maps showing geographcial patterns associated with PDV (1) and AMV (2) and SLP regression maps of geographical patterns associated with PNA (3) and PSA1 (4). The indices are calculated from reanalyses data (ERSSTv5 for AMV and PDV; ERA20C_ERA5 for PNA and PSA1) over the 1900-2014 period using the NCAR CVDP-LE package. Rectangles on the maps approximate the regions over which the indices are defined (see Methods, Sect. 2.1.1) **(b)** Box-plot (or whisker-plot) showing the distribution of Pearson $r$ pattern correlation values along the different historical CMIP6 LEs (between parenthesis on the x-axis is the number of ensemble members within each model). The bottom of every box (color-coded part) shows the first quartile (Q1 or 25th percentile), the top the third quartile (Q3 or 75th percentile) and the horizontal bar between them denotes the median value (Q2 or 50th percentile). The length of the box (from Q1 to Q3) denotes the interquartile range (IQR) while the bottom and upper whiskers (thin lines extending from boxes) extend to the minimum and maximum values, which are calculated as Q1-1.5*IQR and Q3+1.5*IQR, respectively. The black dots are outliers. PNA correlation values are shown in purple, PSA1 in cyan, PDV in red and AMV in green. Yellow boxes show the Mean score denoting the average of the four $r$ values (after applying a Fischer z-transform). White dots denote the mean value.




CESM2 suggests that there are more members with PSA1 patterns resembling the observed ones. The opposite is true for the GISS-E2-1-H model which contains less realizations with similar PSA1 patterns as those from reanalysis. The length of the
cyan box for NorCPM1 indicate that most members fail to represent the spatial patterns of PSA1.

**Table 2.** Pearson $r$ correlations between the simulated (CMIP6 LE) and observed (ERA20C_ERA5, ERSSTv5) spatial patterns of PNA, PSA1, PDV and AMV over the 1900-2014 period. Models are sorted according to the average mean score (column in bold; descending order).

| CMIP6 LE | Mean Score | | | PNA (DJF) | | | PSA1 (ANN) | | | PDV (monthly) | | | AMV (monthly) | | |
|---|---|---|---|---|---|---|---|---|---|---|---|---|---|---|---|
| | min | **mean** | max | min | mean | max | min | mean | max | min | mean | max | min | mean | max |
| UKESM1-0-LL | 0.74 | **0.80** | 0.86 | 0.79 | 0.87 | 0.94 | 0.56 | 0.73 | 0.84 | 0.79 | 0.82 | 0.86 | 0.66 | 0.74 | 0.81 |
| MIROC6 | 0.74 | **0.80** | 0.85 | 0.73 | 0.86 | 0.95 | 0.64 | 0.73 | 0.80 | 0.82 | 0.84 | 0.87 | 0.66 | 0.71 | 0.78 |
| MPI-ESM1-2-LR | 0.74 | **0.79** | 0.83 | 0.73 | 0.84 | 0.93 | 0.65 | 0.77 | 0.82 | 0.75 | 0.80 | 0.84 | 0.63 | 0.71 | 0.78 |
| ACCESS-ESM1-5 | 0.67 | **0.77** | 0.84 | 0.76 | 0.88 | 0.94 | 0.12 | 0.67 | 0.80 | 0.61 | 0.72 | 0.77 | 0.66 | 0.71 | 0.77 |
| CanESM5 | 0.51 | **0.77** | 0.81 | 0.71 | 0.82 | 0.90 | -0.50 | 0.69 | 0.82 | 0.67 | 0.79 | 0.86 | 0.61 | 0.72 | 0.79 |
| IPSL-CM6A-LR | 0.46 | **0.75** | 0.80 | 0.55 | 0.73 | 0.85 | -0.80 | 0.70 | 0.86 | 0.73 | 0.78 | 0.84 | 0.69 | 0.76 | 0.81 |
| CESM2 | 0.59 | **0.74** | 0.84 | 0.83 | 0.88 | 0.92 | -0.67 | 0.23 | 0.82 | 0.82 | 0.86 | 0.88 | 0.68 | 0.72 | 0.78 |
| EC-Earth3 | 0.26 | **0.68** | 0.81 | 0.78 | 0.86 | 0.94 | -0.56 | 0.48 | 0.76 | -0.25 | 0.61 | 0.78 | 0.57 | 0.65 | 0.79 |
| CNRM-ESM2-1 | 0.36 | **0.61** | 0.79 | 0.32 | 0.61 | 0.86 | 0.37 | 0.52 | 0.72 | -0.42 | 0.45 | 0.78 | 0.71 | 0.75 | 0.80 |
| GISS-E2-1-H | 0.41 | **0.60** | 0.79 | 0.63 | 0.80 | 0.90 | -0.72 | -0.06 | 0.74 | 0.66 | 0.77 | 0.82 | 0.60 | 0.68 | 0.75 |
| INM-CM5-0 | 0.41 | **0.54** | 0.63 | 0.53 | 0.65 | 0.74 | -0.31 | 0.28 | 0.66 | 0.47 | 0.51 | 0.56 | 0.57 | 0.65 | 0.71 |
| NorCPM1 | 0.27 | **0.51** | 0.74 | -0.04 | 0.65 | 0.87 | -0.61 | -0.33 | 0.67 | 0.67 | 0.76 | 0.82 | 0.63 | 0.68 | 0.74 |

Along with the release of the CVDP-LE (Phillips et al., 2020), CESM's CVCWG freely distributes results from several CMIP simulations including the CMIP6 1900-2014 historical simulations, from which data used in this analysis have been downloaded. The results include a pattern correlation summary with 11 key spatial metrics of oceanic and atmospheric modes of variability. Similar to the mean score we introduced in the spatial correlation analysis above, the CVDP-LE provides a mean
score averaging the pattern correlations of the 11 metrics used. Although the pattern correlation mean score we calculated is not exactly the same as the one provided by the CVDP-LE tool because the number of indices used is different (four vs 11), the highest-scoring CMIP6 LEs from Table 2 (UKESM1-0-LL, MIROC6 and MPI-ESM1-2-LR) were also the highest scoring ensembles according to the pattern correlation summary provided on the tool's repository (Phillips et al., 2020). Moreover, one simulation from the UKESM1-0-LL ensemble, the r19i1p1f2 realization, was found to obtain the highest mean score based
on both the pattern correlation values published under https://webext.cgd.ucar.edu/Multi-Case/CVDP-LE_repository/CMIP6_ Historical_1900-2014/metrics.html by CVDP-LE authors (Phillips et al. 2020, 0.88 using 11 indices) and our calculations in Table A2 (0.86 using 4 indices)



## 3.2 Regime-oriented causal analysis of observations and reanalyses

Several mechanisms are hypothesized to contribute to PDV and AMV. PDV is initially considered as a mode of internal climate
variability (e.g. Meehl et al., 2021b). However, previous research indicates possible external contributions in the form of solar
(Meehl et al., 2009), greenhouse gas (Meehl et al., 2009; Fang et al., 2014; Dong et al., 2014) or volcanic and anthropogenic
aerosol forcings (Wang et al., 2012; Maher et al., 2015; Smith et al., 2016; Takahashi and Watanabe, 2016). There are studies
suggesting that such external anthropogenic aerosol forcing might be contributing to AMV as well (Booth et al., 2012; Zhang
et al., 2013; Si and Hu, 2017), but evidence from Zhang et al. (2019) supports the notion that the AMV is primarily linked to
internal variability of the Atlantic Meridional Overturning Circulation (AMOC) and its associated meridional heat transport.
This means that the fingerprint of any possible external forcing acting as a confounder is embedded in the timeseries information
of the extracted indices of the modes of climate variability used in this study. The linear detrending we perform prior to
applying PCMCI+ will at least partially reduce such effects. However, as mentioned before, the subtraction of the global mean
temperature for PDV and AMV and the linear detrending of all time series do not address local, nonlinear effects, which could
be related to the aerosol forcing that varies over time and space. It is then important to recall that in this paper, the indices do
not represent a fully isolated internal variability component but rather a mixture of naturally-occurring internal variability and
nonlinear effects of external forcing, mainly in the form of aerosol forcing.

PCMCI+ is applied first to the indices of PDV, AMV, PNA and PSA1 calculated from reanalysis data, as a proxy for
observations, to reveal any causal dependencies between the modes depicted by the observed timeseries information. As it is
assumed that the nature of teleconnections between the different climate variability modes can vary over decadal timescales
depending on the different phases these modes go through, we mask years of data (as discussed in Sect. 2.1.3) to reveal
the causal structures during specific periods (regimes) in time. Reference causal networks obtained by running PCMCI+ on
reanalysis data for the different regimes are shown in Figure 5.

The results show that the causal dependencies (links) between the four modes of climate variability (nodes) change from one
regime to another. Starting from an analysis on the complete period (115 years, upper left panel in Figure 5, and see Table A3
for exact cross-MCI values of the complete period causal graph) PCMCI+ reveals four different links: An 11-year lagged
negative (link arrow is curved and blue) AMV→PDV link (cross-MCI = -0.25) showing that the opposite sign effect on PDV
caused by AMV is lagged by a decade (e.g. positive AMV tends to produce negative PDV about a decade later). Therefore,
this link can be interpreted as lagged opposite sign SST anomaly changes over the Pacific in response to SST anomaly changes
over the Atlantic. The same causal graph features a strong positive (0.53) contemporaneous PDV—PNA link (i.e. link line is
straight) suggesting PDV is strongly associated to PNA. In addition, the complete period graph implies weak South Pacific
teleconnections of both AMV and PDV which are represented by a positive contemporaneous AMV—PSA1 (0.25) link and a
lagged PSA1→PDV link. The latter (PSA1→PDV link) is detected positive at 7 years (0.23) and negative at 15 years (-0.31).
As explained in Sect. 2.1.2, if a lagged link is found at more than one time lag, the causal graph shows the link at the lag when
it is most significant (i.e higher absolute cross-MCI value) and labels the other time lags after a comma ( $|-0.31|$ vs $|0.23|$ in
this case, thus the "15, 7" label on the PSA1→PDV link; see upper left panel in Figure 5).




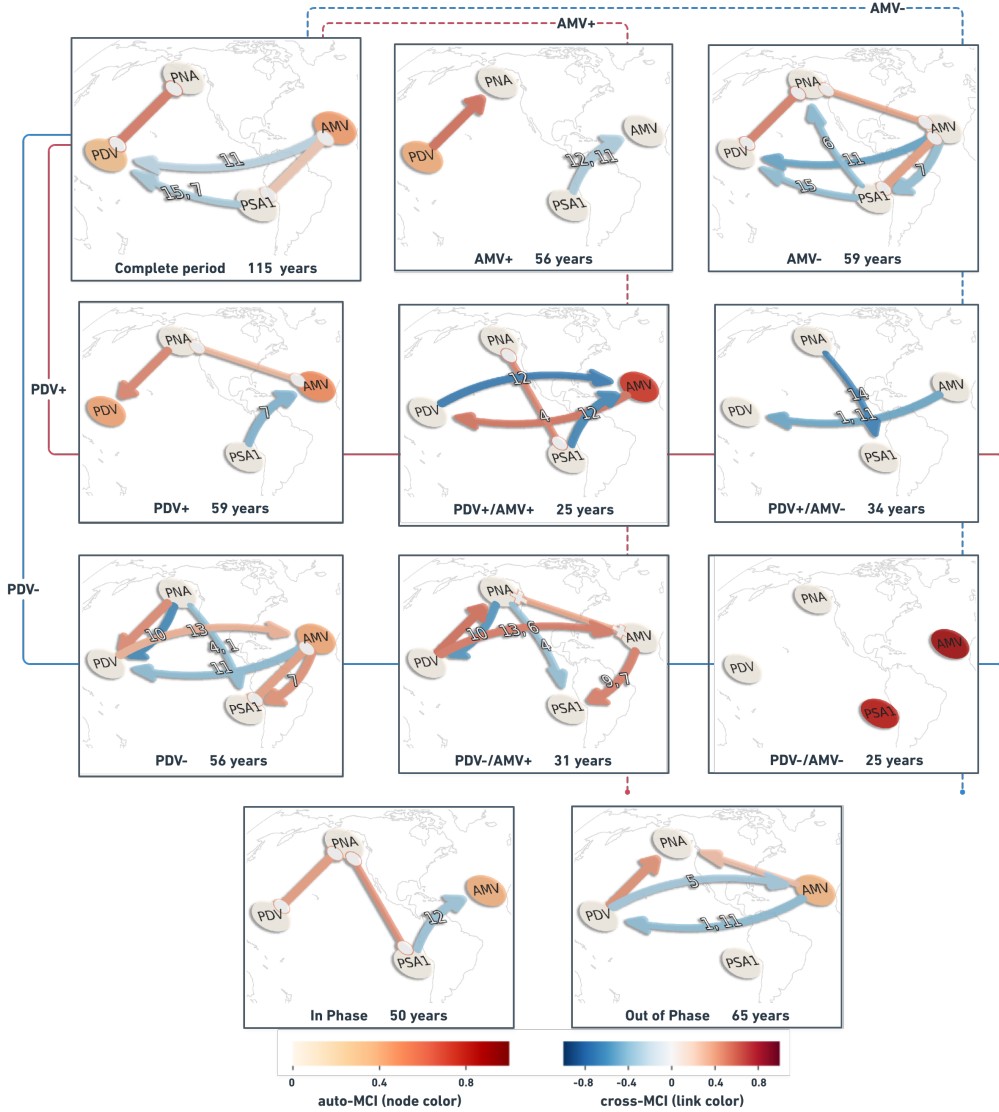

**Figure 5.** Causal networks calculated with PCMCI+ from reanalysis data for the complete 1900-2014 period (upper left panel) and the different regimes. Nodes represent the timeseries associated with each climate variability index (see node labels) masked according to the predefined regimes. Node colours indicate the level of autocorrelation (auto-MCI) as the self-links of each node with darker red indicating stronger autocorrelations (color bar at lower left) while the color of the arrows (termed "links" here) denotes the inter-dependency strength (cross-MCI) with blue indicating opposite-sign (or negative) inter-dependency and red indicating same-sign (or positive) inter-dependency strength (color bar at lower right). Small labels on the curved links indicate the link-associated time lags (unit = 1 year). Straight links show contemporaneous inter-dependencies happening with no time lag (i.e. $\tau < 1$). Each network is sub-labeled with its respective regime name and the total number of unmasked years (time-steps) that characterize that regime (label and number of years at bottom of each panel). Lines going through the panels are to help visualize which combinations make up the regimes. Solid lines are for PDV, dashed for AMV. Red for warm (+), blue for cold (-) phases (e.g. PDV+/AMV- regime panel has a solid red line and dashed blue line going through it).




The complete period graph in the upper left in Figure 5 is useful to show the causal dependencies happening throughout the whole observational record used. However, this methodology can also be used to look at specific regimes to notice the change in dependencies arising from the physical state of the Atlantic and Pacific basins during those time periods. For example, the

causal graphs from PDV+ and PDV- regimes indicate that direct decadal AMV—PDV interactions occur only during the PDV-regime (third row, left panel in Figure 5), whereas during the PDV+ regime (second row, left panel in Figure 5) we find a contemporaneous atmospheric teleconnection from PNA to both AMV and PDV. This difference could be explained by the fact that the PDV- regime comprises two important Atlantic variability events: the 1920s AMV phase-switch from negative to positive (see dashed lines showing low-pass AMV in Figure 3a) and the subsequent switch from positive back to negative

during late 1960s.

The regime-oriented nature of this causal analysis provides for a separation of signals, for example delineating the PDV+ regime that depends on the AMV phase during those 59 years (second row panels in Figure 5). The short length of timeseries, in addition to the time-varying aerosol forcing during such regimes, can lead to inconclusive causal estimations. The PDV-/AMV-panel at the right of the third row in Figure 5 (25 years) shows strongly auto-correlated AMV and PSA1 patterns but no apparent

links between any of the four variables. However, these short regimes might also reveal interesting causal relations that are not apparent when analysing longer periods. This is the case for the causal graph from the 25 years of the PDV+/AMV+ regime (central panel in Figure 5), which is the only one to feature a strong negative PDV→AMV link and a positive AMV→PDV link with comparable strength. Since the causal parents that drive the variables (other variables or the same one at different past time-steps) can originate from a masked period with respect to $\tau_{max}$, it implies, for example, that the strong 12-year lagged

negative PDV→AMV causal link estimated during the PDV+/AMV+ regime (second row, central panel in Figure 5), might have fingerprints originating from a previous regime.

The limitation presented by the length of unmasked timeseries during specific short regimes is eliminated when combining them. For this reason, we show in the bottom of Figure 5 causal graphs for In-Phase and Out-of-Phase regimes. We detect the negative lagged direct AMV→PDV and PDV→AMV only during the Out-of-Phase regime with a strong positive extra-

tropical PDV→PNA teleconnection and a weaker AMV→PNA teleconnection. The In-Phase regime features a fast (zero lag) PDV teleconnection to PNA, PNA connection to PSA1, and a 12-year lagged PSA1→AMV link. As finite sample errors can lead to false positives as well as false negatives (missing links), it is difficult to attribute a physical explanation to every detected link. Though here both are thought to be driven by tropical precipitation and heating anomalies, we refrain from assigning any processes that might be behind the direct PNA—PSA1 causal links due the lack of knowledge regarding possible direct links

between the North Pacific and South Pacific extra-tropics.

Through observations of the long-term variability patterns and pacemaker simulations of Atlantic and Pacific ocean basins, Meehl et al. (2021a) explain how positive AMV could produce an opposite-sign response, mainly through the atmospheric Walker circulation, leading to negative PDV, and then the negative PDV subsequently contributing a same-sign response in the Atlantic driving the AMV from positive to negative phase. This mutual contrasting response from one basin to the other can

be interpreted through the blue (negative cross-MCI) lagged AMV→PDV links and the reddish (positive cross-MCI) lagged PDV→AMV links in the causal networks in Figure 5. The results in Figure 5 show that the lagged AMV→PDV causal link





has been estimated over the complete period and during five out of the 10 regimes (AMV-, PDV-, PDV+/AMV+, PDV+/AMV-, Out-of-Phase). During four of these regimes, the link can be interpreted as a lagged opposite-sign effect of AMV on PDV (blue curved link). The study of Meehl et al. (2021a) suggests that in addition to the tropical Walker circulation, positive
convective heating and precipitation anomalies in the tropical Pacific establish extra-tropical teleconnections to PNA and PSA which contribute to the same-sign effect of PDV on AMV. The causal graph from the 31 years of the PDV-/AMV+ regime (third row, middle panel in Figure 5) shows two possible pathways for this same-sign effect of PDV on AMV. During that regime, PCMCI+ estimates a strong positive 13- and 6-year lagged PDV→AMV link (the 13-year lagged link was also found during the 56 years of the PDV- regime) but also shows a positive PDV→PNA—AMV contemporaneous teleconnection where
PNA seems to mediate the same-sign effect of PDV on AMV. Therefore, this analysis presents additional evidence that AMV (although potentially affected by a forced aerosol signal) might serve as a predictor of decadal variability over the Pacific (hence for PDV) and eventually the other way around (d'Orgeville and Peltier, 2007; Zhang and Delworth, 2007; Chikamoto et al., 2015; Johnson et al., 2020).

     An earlier study from Zhang and Delworth (2007) proposed a mechanism in which positive (negative) AMO would lead to
high (low) SLP anomalies over the North Pacific and eventually a positive (negative) PNA pattern. This weakening (strengthening) of the Aleutian low associated with the PNA pattern projects onto the multidecadal mode of variability over the North Pacific. The response of North Pacific SST to the anomalous PNA pattern induced by AMO is hypothesised to be lagged due to Rossby wave propagation and gyre adjustment where the authors found a 3-year lag when using a model simulation compared to a 12-year lag when they analysed the observed pattern. The extra-tropical contributions of PNA and PSA1 on the mutual
PDV—AMV interactions can be concluded from causal graphs constructed during different regimes (see Figure 5). AMV-, PDV+, PDV-/AMV+ and Out-of-Phase are all regimes that suggest mutual Atlantic-Pacific connections can be established via PNA. The causal networks from the complete period and AMV- regime show that these inter-basin connections can also happen through PSA1.

     Previous research also showed that components of the PDV can be forced by tropical Pacific variability and/or driven by
atmospheric stochastic forcing which are both closely tied to Aleutian low variability associated with the PNA pattern (Newman et al., 2016; Johnson et al., 2020). This literature finding on the PDV—PNA teleconnection validates the contemporaneous PDV—PNA causal link estimated by PCMCI+ during most regimes (all except PDV+/AMV+ and PDV-/AMV-; see causal networks in Figure 5) with a strong positive cross-MCI value. The link is directed in some regimes (straight links with arrowhead, e.g. during PDV+ regime) while it is unoriented during other regimes (straight links with no arrowheads, e.g. during AMV-
regime). A 10-year lagged negative PNA→PDV link appears during the PDV- regime in Figure 5 (and during PDV-/AMV+) which suggests that an extra-tropical teleconnection to PNA might have the opposite effect during longer time lags.

     Generally, lags ranging from interannual (1 to 5 years, Wu et al., 2011; Meehl et al., 2021a) to decadal (12 to 17 years, Wu et al., 2011; Chylek et al., 2014) timescales have been proposed by previous studies for Atlantic-Pacific interactions which fall in the same range of time lags at which causal links have been estimated by PCMCI+ in this study.

To further justify the credibility of the constructed causal networks, we use the estimated causal graph from the complete period to construct a model that explains the lagged correlation structure of the reanalysis dataset. This is done by fitting a linear





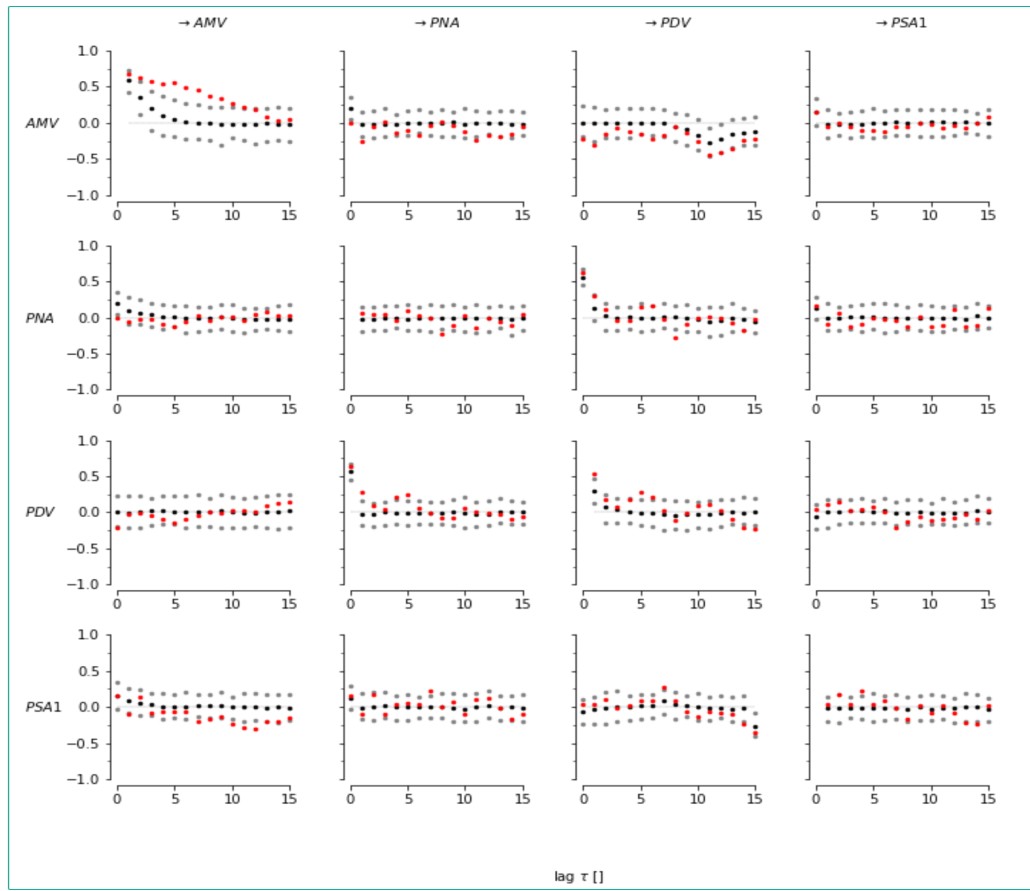

**Figure 6.** Lagged correlations of original data (complete period graph from reanalysis data) in red shown together with lagged correlations of an ensemble of synthetic data generated by a linear Gaussian structural causal model with causal coefficients and noise structure estimated from the original data. The mean lagged correlations from the synthetically generated data are shown in black, and their 5th-95th percentile range in grey.

structural causal model to causal parents taken from the original reanalysis causal graph. We generate 100 realizations following a linear Gaussian causal model with the noise structure estimated from the noise covariance matrix of residuals. Figure 6 shows lagged correlations of the original data in red with the mean lagged correlations from the synthetically generated data in black and their 5th-95th percentile range in grey. The original lagged correlations (red) fall mostly within the 90% range (with the clear exception of AMV's lagged auto-correlation at the upper left of Figure 6). This means that a linear Gaussian model with the same links as those from the reconstructed causal graph can well explain the whole lagged correlation structure of the original data for PNA, PDV and PSA1. Such a lagged-correlation matrix (Figure 6) also unveils how the dependencies between different variables change over time.





### 3.3 Regime-oriented causal model evaluation of the CMIP6 Large Ensembles

With the overall high level of fidelity that several models show in simulating the spatial patterns of at least the major modes of climate variability presented in this paper (see Figure 4b), it is crucial to test whether these simulations also account for the possible lagged causal pathways between these different modes. To benchmark the dependency structures in model simulations, the simulated causal graphs are compared to those from reanalysis datasets (ERSSTv5 for PDV and AMV, ERA20C_ERA5 for PNA and PSA1). The constructed causal graphs from the previous section illustrate the connections occurring between the different modes of climate variability during different regimes, as estimated from reanalysis data. Relative to reanalysis, we consider the causal graphs from Figure 5 as reference for the CMIP6 model evaluation to be demonstrated in this section.

The exact same PCMCI+ setting (see Methods; Sect. 2.1.2 and Sect. 2.1.3) used in the section above is applied for timeseries indices calculated from every realization of the CMIP6 models listed in Table 1. In Sect. 3.1, we found that, overall, the spatial patterns of these simulated indices compare fairly well to the observed ones (Figure 4, Table 1). The purpose of this section is to show how the causal fingerprints in these simulations compare to those observed. For every realization, the analysis is run for the complete period in addition to the 10 different regimes, similar to the regime-oriented setting on reanalysis data in the section above. As the PDV and AMV phases occur in model simulations at different time periods than those in reanalysis (due to randomly generated internal variability and time-varying forcing caused mainly by aerosols), models need not show similar networks for the same periods as in observations. However, we can assess the degree of similarity in the causal fingerprint that these simulations hold within their modeled dynamics. The results of every realization during every regime are compared to the reference networks from reanalysis data during that regime.

To illustrate results from an individual model, we aggregate causal networks from 65 realizations from the CanESM5 model in Figure 7. This figure shows networks with links of variable thickness indicating that some links are found in most ensemble members during that specific regime (thick links, e.g. PDV—PNA in most regimes) compared to other links (thinner links, e.g. PDV→AMV in most regimes) which were detected only by a small fraction of ensemble members. The thicker the link, the more agreement between members of the same ensemble in detecting that specific link. We also label the links with the rounded mean lag at which they are detected in the ensemble members. The link color in this ensemble summary (Figure 7) is informative of the level of agreement between ensemble members in estimating that causal link with the same sign. The clearer the shade of blue (negative) or red (positive), the better agreement between ensemble members in simulating the link with the same sign. For example, the color of AMV—PNA links in most regimes (although mostly estimated by few members during each regime, i.e. relatively thin links; see Figure 7) tend towards reddish shades suggesting that the CanESM5 members, in which such links were estimated, agree that the causal link is of positive sign. This can be translated to the positive (negative) AMV driving positive (negative) PNA and vice versa. This can be seen on all causal networks in Figure 7, except the ones from PDV+/AMV+ and PDV-/AMV+ regimes indicating that in a few of the CanESM5 realizations, AMV would induce an opposite sign response on PNA (see thin blue AMV→PNA links on PDV+/AMV+ and PDV-/AMV+ causal graphs in Figure 7).

Other than the PDV—PNA links (estimated by most ensemble members during all regimes), the occurrence of a link in the CanESM5 model seems to vary from one regime to another. This is less true for the complete period, the In-Phase and the





**Figure 7.** Ensemble summary of the CanESM5 LE model. Similar to Figure 5, but aggregating causal networks from 65 realizations. The link width here shows the fraction of ensemble members that feature that link relative to the total ensemble size (here 65); i.e. the thicker the link, the more ensemble members were found to estimate it during that specific regime. Link colors here translate the mean cross-MCI value among the ensemble members that estimated such link (color bar at lower left). Links of very light color are those that ensemble members agree little on their partial correlation sign. The link labels indicate the average time lag (rounded to the nearest integer) at which the link is estimated among the fraction of ensemble members that find such link.





Out-of-Phase regimes. The complete period ensemble causal graph distinctly shows AMV—PNA interactions as same-sign

causal links between the two modes. The same graph (upper left panel in Figure 7) also shows a clear blue AMV→PDV link, demonstrating the opposite-sign response driven by AMV on PDV, similar to the one featuring in the complete period causal graph from reanalysis data (upper left in Figure 5). The color and width (thickness) of this AMV→PDV link in the complete period graph in Figure 7 (upper left panel) suggest that the link was estimated with negative cross-MCI values by a considerable fraction of CanESM5 simulations.

A more evident network similarity is evinced during the Out-of-Phase regime. Both the graph from reanalysis (Figure 5, Out-of-Phase) and the CanESM5 ensemble graph (Figure 7, Out-of-Phase) display a short lagged (1-year lag and 2-year mean lag, respectively) opposite-sign (blue, negative cross-MCI) AMV→PDV causal link. Moreover, the two graphs (Out-of-Phase causal networks in Figure 5 and Figure 7) suggest a same-sign (red, positive cross-MCI value) contemporaneous and short-lagged (1 year) PDV—PNA causal links, and weaker same-sign (lower positive cross-MCI values) AMV—PNA links. The

latter links are instantaneous in the reanalysis data but lagged in CanESM5. However, the short mean lag (2 years) in the simulated CanESM5 Out-of-Phase graphs imply that several members estimate a contemporaneous link.

The CanESM5 ensemble causal graph during the In-Phase regime at the bottom of Figure 7 demonstrates the advantage of using LEs. While the reanalysis graph during this regime suggests only PDV—PNA and lagged PSA1→AMV teleconnections (with a debatable contemporaneous PNA—PSA1 link), the CanESM5 ensemble graph displays a clear same-sign

lagged AMV→PDV link with a third of its ensemble members simulating such a dependency. Despite the fact that the positive AMV→PDV link is not detected in reanalysis during the In-Phase regime (Figure 5, In-Phase regime causal graph), literature supports this contrasting effect estimated by CanESM5 model data (Wu et al., 2011; Meehl et al., 2021a). Model simulations can therefore explain causal dynamics between modes of climate variability that might not definitively appear when analysing observations. There is less doubt about the agreement between members of the CanESM5 ensemble, and also when compared

to reanalysis, about the occurrence of an AMV→PDV link with an opposite sign during the Out-of-Phase regime.

Ensemble summary plots are calculated for all CMIP6 LEs from Table 1 but we only chose to display them for CanESM5 in Figure 7. The ensemble summary of causal networks from reanalysis data and the 12 CMIP6 models for the complete 1900-2014 period, Out-of-phase and In-Phase regimes are shown in appendix Figs. A1-A3 respectively. In order to measure the level of similarity between observed and individual ensemble member networks across all the CMIP6 models, $F_1$-scores are

computed for every realisation and every regime. The results reveal that most CMIP6 Large Ensembles show better network (causal graph) similarity with reanalysis reference networks during the Out-of-Phase regime, compared to the networks drawn during the other regimes and/or the complete period. The whisker plot in Figure 8a shows the distribution of $F_1$-scores across the CMIP6 LEs for the complete period (light blue boxes), the In-Phase regime (dark blue) and the Out-of-Phase regime (green). The range of scores during the other regimes (not shown) was found to be much lower compared to the scores during

the regimes shown in Figure 8a. The white scatter points show that on average, CESM2, CanESM5, MIROC6 and MPI-ESM1-2-LR LEs clearly display better network similarity with observations during the Out-of-Phase regime. The highest scores during this regime (0.92) belong to members of CanESM5 and MIROC6 LEs (see location markers on whisker plot). Figure 8b compares Out-of-Phase causal graphs from these highest-scoring realizations (and their low-pass filtered AMV,





PDV timeseries) to those from reanalysis. The networks in Figure 8b agree on the 1-year lagged AMV→PDV link. The
positive contemporaneous PDV—PNA link is directed differently in reanalysis and CanESM5 r11i1p2f1, but unoriented in
CanESM5 r17i1p2f1 and MIROC6 r20i1p1f1. The Out-of-Phase graphs from these realizations also agree on a same-sign
contemporaneous AMV—PNA dependency, with a lower cross-MCI value (weaker) than that of the PDV—PNA connection.

In Figure 9 we perform intra- and cross-model network comparisons for the complete period and long regimes. This is
done by computing $F_1$-scores with every single realization as a reference. Averaging the $F_1$-scores by ensemble produces an
$F_1$-matrix for every regime in the form of heat maps translating the degree of similarity (the redder the color, the greater the
similarity) in causal dynamics between members of the same LE (boxes on the main diagonal) and pairwise causal similarity
between different LEs (boxes outside the main diagonal). Every grid box on the heat maps shows how the corresponding
CMIP6 model from the axis on top (see model names on x-axis top of every panel) compares to the reference corresponding
CMIP6 model (see model names on y-axis left of each panel). We exclude the short regimes (PDV+/AMV+, PDV+/AMV-,
PDV-/AMV+ and PDV-/AMV-) from this comparison as the PCMCI+ results during these regimes tend to be inconclusive
(i.e. the regimes are too short to estimate any causal link for several simulations from different models). The heat maps show
that CNRM-ESM2-1 LE clearly stands out as the most dissimilar model during most regimes. This is seen in the third row
and third column (from top to bottom, left to right) of each heat map ($F_1$-matrix of every regime) in Figure 9 indicating the
lowest $F_1$-scores (yellow and white lines on the heat maps; see also color-bar). The model doesn't only have the lowest level
of agreement with other ensembles but also shows poor accordance within its own members. Generally, the other CMIP6
models exhibit better network similarity during longer regimes (Complete period, AMV+, AMV-, PDV+, PDV-, In-Phase,
Out-of-Phase). Members of CESM2 LE strongly agree between each other in terms of causal fingerprints displayed during
the analysis on the complete period; this is shown by the dark red box on the second row and second column of the complete
period heat map ($F_1$-matrix). The INM-CM5-0 LE shows low average $F_1$-scores during the PDV+ and Out-of-Phase regimes,
but it surprisingly shows the most agreement between its own ensemble realizations during the complete period, AMV-, PDV-
and the In-Phase regimes (see dark red grid boxes in the center of heat maps of these regimes on Figure 9). This implies that
the INM-CM5-0 ensemble might involve mostly simulations where PDV and AMV are in the same phase.

The skill of CESM2, CanESM5, MIROC6 and MPI-ESM1-2-LR in recreating the observed causal pathways of the Out-
of-Phase regime is also manifested through the better similarity the members of these models show when compared to each
other. The heat maps ($F_1$-matrices in Figure 9) serve to distinguish models with similar causal dynamics. The specified range of
internal variability within realizations of the same ensemble (combined with the model-simulated time-varying aerosol forcing)
can also be inferred by comparing one LE to itself.

## 3.4 Discussion

Previous research already suggested the improvement in simulation of dominant modes of climate variability throughout the
different phases of the CMIP archive (Fasullo et al., 2020; Eyring et al., 2021). Although, in general, models are able to capture
the spatial patterns of these modes, CMIP6 revealed discrepancies in the skill these LE simulations display when recreating
the observed modes. Some models perform very well, while there is still room for improvement for others. This conclusion



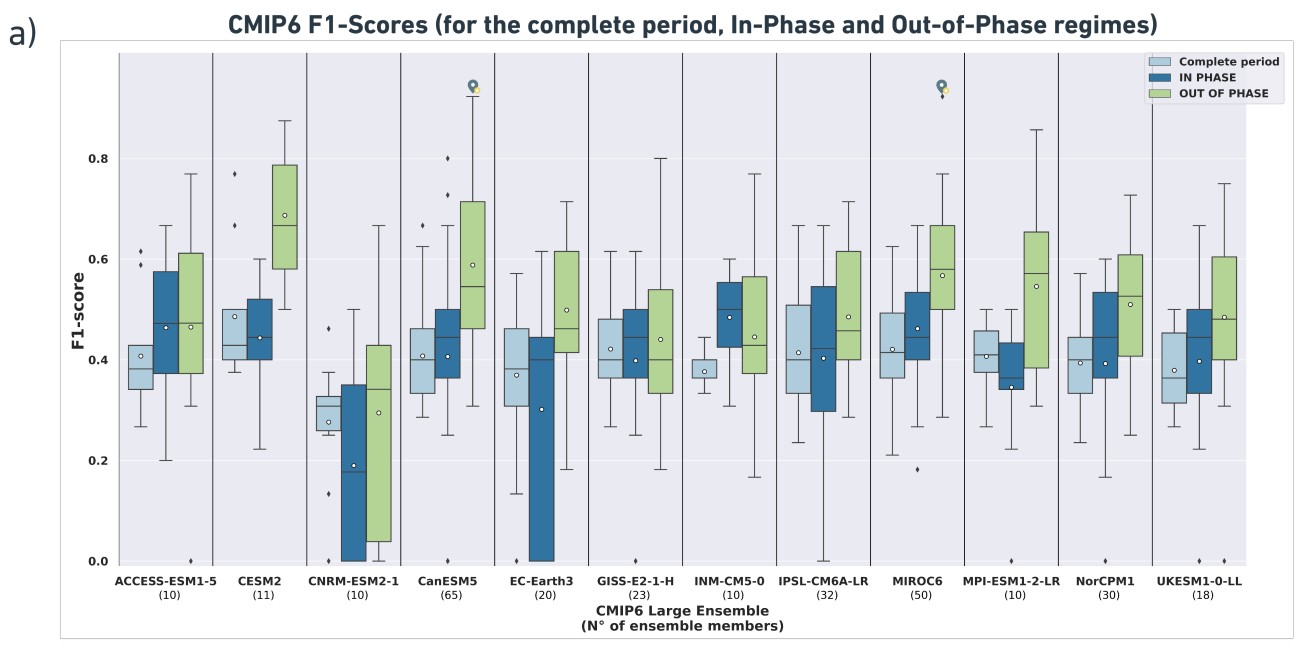

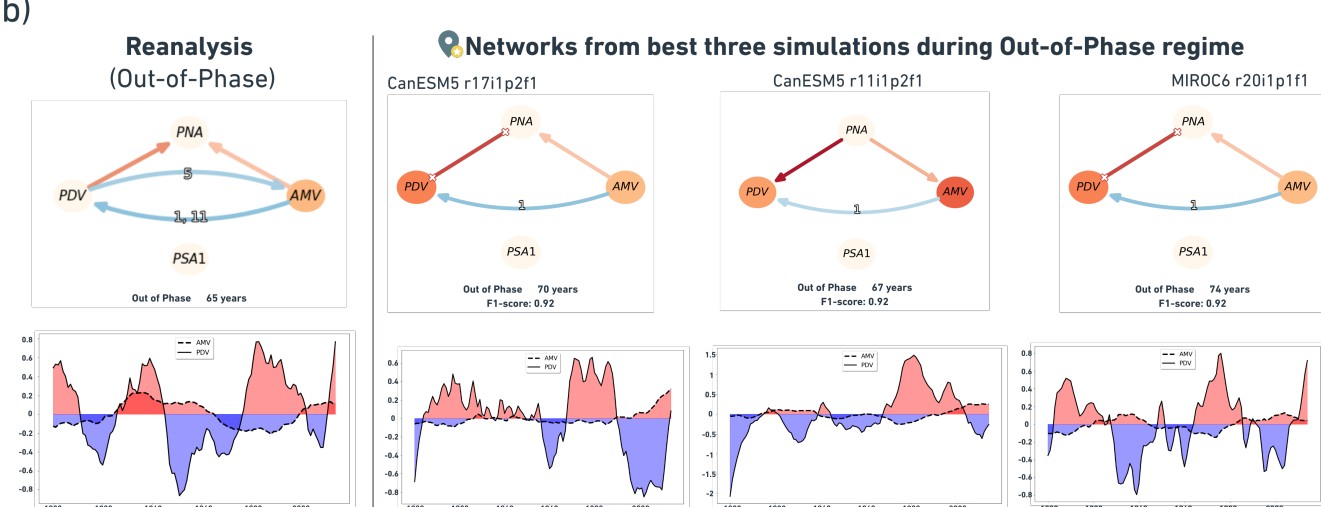

**Figure 8. (a)** Whisker plot showing the distribution of $F_1$-scores across the CMIP6 LEs for the causal analysis for: the complete period (light blue boxes), the In-Phase regime (dark blue boxes) and the Out-of-Phase regime (green boxes). White scatter points denote the mean LE $F_1$-scores. **(b)** Reference causal network estimated from reanalysis during the Out-of-Phase regime (left, with low-pass AMV and PDV timeseries below) compared to networks and timeseries from three CMIP6 simulations (right, with simulated low-pass AMV and PDV timeseries below each network) with the best network similarity i.e. highest $F_1$-score.





**Figure 9.** Matrices of average $F_1$-scores for pair-wise network comparisons between ensemble members of 12 CMIP6 LEs during every regime. Boxes on the main diagonal translate the level of similarity between members of a single CMIP6 ensemble. Boxes outside the main diagonal show the similarity between realizations of a CMIP6 LE compared to realizations from another CMIP6 LE (taking every realization as a reference at a time, before averaging across every LE). The redder the grid box, the better causal network similarity it translates when comparing realizations of the corresponding CMIP6 model (x-axis coordinate name on top of each panel) to causal networks from the corresponding reference CMIP6 model (y-axis coordinate on the left of each panel). The matrices for the short regimes (PDV+/AMV+, PDV+/AMV-, PDV-/AMV+ and PDV-/AMV-) are not shown as their results are not conclusive since PCMCI+ fails to estimate any causal networks for several members of different ensembles.





is illustrated through results of the pattern correlations in Sect. 3.1 and the wide range of comparison metrics produced and published by the CVDP-LE authors (Phillips et al., 2020). The ability of CMIP6 LEs to recreate the spatial patterns of modes of climate variability does not, however, ensure that they simulate the connections between those modes. Relative to the reference networks from reanalysis datasets during the Out-of-Phase regime, CESM2, CanESM5, MIROC6 and MPI-ESM1-2-LR LEs display the highest degree of similarity. These CMIP6 LEs were found to also simulate most spatial patterns with high correlation coefficients. On the other hand, other LEs such as the UKESM1-0-LL and ACCESS-ESM1-5, despite their high correlation with the observed spatial patterns, do not exhibit the same level of similarity when comparing their causal networks to the reference networks. This discrepancy might be due to the difference in external time-varying aerosol forcing with respect to random internally-generated variability.

In Figure 10 we plot the $F_1$-scores for all realizations (color- and marker-coded by CMIP6 ensemble, see legend) for the long regimes with respect to the mean-score of $r$ spatial correlations from Sect. 3.1. Similar to Figure 9, we choose not to show the scatter plots for the short regimes. As the mean-scores of spatial correlations are the same for all regimes (computed between the regression maps on the whole 1900-2014 timeseries of the indices), how high (low) a single scatter point can get during a certain regime reveals its causal network similarity (dissimilarity) with reanalysis during that regime. The scatter points closer to the top right corner of each plot belong to realizations which simulate better the spatial patterns and causal fingerprints of reanalysis. Considering only the complete period panel (upper left in Figure 10), the upper right corner of this panel shows mainly realizations from CESM2 (orange crosses), MIROC6 (yellow triangles), CanESM5 (red plus signs) models. From the same panel, we can notice, for example, that the UKESM1-0-LL realizations (orange 5-pointed stars) have great spatial pattern similarity with reanalysis. These UKESM1-0-LL realizations, however, do not show high similarity when comparing their causal fingerprint to that concluded from reanalysis data. The same can be said about MPI-ESM1-2-LR realizations (cyan 6-pointed stars) which, in spite of their high level of skill in recreating the spatial regression patterns of the four modes of climate variability, fail to obtain $F_1$-scores as high as those from CESM2, CanESM5 or MIROC6 during most regimes. Only during the AMV+ and Out-of-Phase regimes that very few MPI-ESM1-2-LR simulations exceed the 0.7 $F_1$-score bar. Overall, we can conclude that CESM2 (orange crosses), CanESM5 (red plus signs) and MIROC6 (yellow triangles) undoubtedly outperform other LEs in this evaluation. This is proven through the consistency that simulations from these two LEs show in resembling the observed causal fingerprints during the different regimes. Despite obtaining high spatial correlation coefficients, two members of the IPSL-CM6A-LR model (grey scatters) show the best network similarity with reanalysis during the PDV+ regime while three other members of this model show no similarity during the same regime.

It is worth mentioning that the number of realisations within an LE appears to increase the chance of a model to comprise a simulation with similar dependency structures as those found in observations. The three simulations with the highest $F_1$-scores during the Out-of-Phase regime (see Figure 8) belong to either CanESM5 or MIROC6 which are the LEs with the highest number of realisations (65 and 50 ensemble members, respectively). This is likely related to the number of realizations needed to capture similar random internal variability to the one observed in reanalysis data. This is less valid for the CESM2 model, which with only 11 realizations, contains simulations with high $F_1$-scores during most regimes shown in Figure 10. In general, modeling centers previously contributed only a small number of realizations to international climate change projection







**Figure 10.** Scatter plots: $R_{coef}$ mean score (spatial correlation with reanalysis, x-axis) vs $F_1$-score (network similarity with respect to reanalysis, y-axis) during the different regimes. Spatial correlation values do not change from one regime to another; these are the same mean scores calculated from the Pearson $r$ coefficients of the four modes in Sect. 3.1 over the 1900-2014 period. Similar to Figure 9, scatter plots are shown only for the long regimes.



assessments [e.g., phase 5 of the CMIP (CMIP5; Taylor et al., 2012)]. As a result, model-associated errors and internal climate variability remained difficult, if not impossible, to disentangle (Kay et al., 2015). In this paper, as CMIP6 includes LE models,

we overcome this sampling problem by using at least 10 realizations per model (see Table 1). In this way, we have a better estimate of the natural internal variability and the externally forced part. The larger the ensemble size, the more likely that the observed internal variability falls within the plausible internal variability range simulated by that particular LE model realizations.

    The spatial pattern correlation analysis (Figure 4), the resulting $F_1$-scores with respect to reanalysis (Figure 8a), and the

CMIP6 pair-wise network comparisons (Figure 9) call for the need to investigate the coupling attributes and the simulated internal variability in the CNRM-ESM2-1 ensemble, as its realizations clearly fail to reproduce the observed spatial patterns and causal links between modes of climate variability compared to other CMIP6 LEs. The relatively large distribution of spatial correlation values for the simulated PNA and PDV modes (see purple and red boxes of CNRM-ESM2-1 in Figure 4b), suggest spatial disagreement between the realizations of CNRM-ESM2-1 model regarding the expressed PNA and PDV patterns. This

might be the result of a relatively large distribution of forced PNA and/or PDV trends. This can be supported by the timeseries metrics provided by CVDP-LE which reveal that, among the models analysed in this paper (see Table 1), CNRM-ESM2-1 holds the largest $10^{th}$-to-$90^{th}$ percentile range of linear PDV trends (-0.89 per 115 years to 1.18 per 115 years) during the 1900-2014 period. These values can be found on the PDV timeseries ensemble summary figure (https://webext.cgd.ucar.edu/Multi-Case/ CVDP-LE_repository/CMIP6_Historical_1900-2014/pdv_timeseries_mon.summary.png), as part of the historical 1900-2014

CMIP6 variability diagnostic results distributed by the CVDP-LE authors (Phillips et al., 2020). Considering that the model only counts 10 realizations, the large $10^{th}$-to-$90^{th}$ percentile range reveals that the forced PDV trend can be significantly different from one CNRM-ESM2-1 simulation to another. This translates not only to the dissimilarity in terms of spatial PDV patterns within the ensemble members but most probably leads to very different causal dynamics too. The latter can be seen through the causal networks in Appendix Figure A1 and Figure A2 where CNRM-ESM2-1 simulations hardly agree on the

sign of the PDV—PNA links (appearing with lighter shades of red) compared to the other CMIP6 models (where PDV—PNA links appear with darker shades of red).

    In the present work we defined regimes explicitly based on the phases of PDV and AMV. There are also methods to agnostically extract underlying regimes and their corresponding causal graphs from timeseries (Saggioro et al., 2020), but these are not reliable for the small sample sizes in combination with the high dimensionality of the present datasets.

## 4    Summary

Applying PCMCI+ to reanalysis data revealed that the direct decadal opposite-sign response from AMV to PDV, described by Meehl et al. (2021a) occurs not only during the analysis of the complete 1900-2014 period (with 11-year time lag), but also during several specified regimes: PDV- (11-year lag), AMV- (11-year lag), PDV+/AMV- (1- and 11-year lags), and when PDV and AMV are out of phase (1- and 11-year lags). These regimes vary from 34-year long (for PDV+/AMV-) to 65-year

long (for Out-of-Phase). For the shorter PDV+/AMV+ regime (25-year long) we detect a positive same-sign response from





AMV to PDV with a 4-year time delay. The causal networks constructed from the reanalyses datasets have also revealed the same-sign response from PDV to AMV during two regimes: PDV- (59-year long) and PDV-/AMV+ (31-years long). In other words, the regime-oriented causal analysis indicates that AMV might serve as an early predictor of decadal variability over the Pacific. We also find an indirect connection between the Atlantic and Pacific, which is established via PNA during AMV-

and PDV+ regimes (both 59-year long), and during PDV-/AMV+. The latter is one of the two regimes that feature a same-sign response from PDV to AMV. An indirect connection between Atlantic and Pacific via the Pacific–South American Pattern is found during the complete 1900-2014 period, where AMV is positively linked with PSA1, but PSA1 has a negative lagged link to PDV. During AMV- regime, the causal graph shows opposite-sign AMV→PSA1→PDV lagged connections.

As an example for the regime-oriented causal analysis on CMIP6 models, we showed the CanESM5 ensemble averaged

causal graphs which indicate that the opposite sign effect of AMV on PDV (blue AMV→PDV link) is recreated by several realizations (38 out of 65) during the Out-of-Phase regime, agreeing with the reanalysis results and literature findings (Newman et al., 2016; Johnson et al., 2020). Appendix Figure A1 and Figure A2 show that this opposite sign lagged effect of AMV on PDV was clearly present in simulations belonging to CESM2 and MIROC6 ensembles (AMV→PDV links are clearly blue). The PDV teleconnection to PNA in the form of mutual same-sign response (positive cross-MCI links) was clearly present in

most realizations of not only the CanESM5 model (Figure 7) but most of the CMIP6 LE simulations analysed. This is true considering the exception of the CNRM-ESM2-1 simulations which show less agreement between each other on the sign of the PDV—PNA links (appearing with lighter shades of red in Figure A1 and Figure A2) compared to the other CMIP6 models.

The evaluation of the Large Ensembles from the CMIP6 archive presented in this paper unveiled how a model performs compared to other models in terms of simulating observed spatial patterns and causal pathways between modes of climate

variability. Most CMIP6 models were found to score better during the Out-of-Phase regime, with CESM2, CanESM5, MIROC6 and MPI-ESM1-2-LR as the best performers during this regime. We showed the importance of using LEs in causal model evaluation to address the sampling issue and explained possible causal pathways during specific regimes that might not appear in causal networks constructed from reanalysis data. Several CanESM5 realizations suggested a same-sign AMV→PDV link during the In-Phase regime. This link did not appear on the In-Phase regime causal graph reconstructed from reanalysis. This

same sign response is nonetheless documented by previous research (Wu et al., 2011; Meehl et al., 2021a). The CanESM5 and MIROC6 models with the highest numbers of members were found to outperform other models in simulating observed causal patterns during the long regimes (see Figure 8a). Interestingly, the CESM2 model, with a relatively smaller ensemble size (11 realizations), was also found to display larger causal fingerprint similarity with reanalysis during the long regimes. The causal network similarity between different CMIP6 LE models was also assessed throughout this paper. Simulations from CESM2,

CanESM5 and MIROC6 models also largely resemble each other and those from the MPI-ESM1-2-LR model in terms of estimated causal networks during most regimes (Figure 9).

A deepened intra-model comparison remains essential to evaluate how realizations of the same model ensemble differ from one another. The 'ripf' identifier of every simulation within the CMIP6 LEs used in this study show that some LEs only include realizations (r) with the same initialization (i), physics (p) and forcing (f), while other LEs contain realisations with different



physics or forcing. On that account, it is of high importance to inspect the documentation provided by modeling groups on the relevant realization attributes of their model ensemble.

Causal model evaluation is also helping to better understand remote contributions to internal variability over specific regions. As we are not subtracting the ensemble mean (representing the forced response), the causal links found when analysing observational reanalysis and CMIP6 historical simulations are thus expected to include external forcing contributions, especially

those from space and time-varying aerosol radiative forcing. It is therefore crucial to separate the internal variability component from the externally forced part to gain a better understanding on the effects of external forcings on Atlantic-Pacific interactions. Meehl et al. (2021a) recently examined this effect through timeseries pacemaker experiments in which effects from aerosols are removed (by fixing aerosols at 1920 values). The approach and findings presented here motivate a follow-up study where pacemaker, pre-industrial control, and future scenario simulations are to be analysed through causal discovery algorithms to

reveal the impact of climate change on the teleconnections and interactions between major modes of climate variability. Overall, the regime-oriented causal model evaluation followed in this study has the potential of a powerful methodology that can be applied in a number of environment-related topics, offering tremendous insight to improve the understanding of the complex earth system and the state-of-the-art of climate modeling.

*Code and data availability.* The complete CVDP-LE diagnostic for the 1900-2014 historical run can be found on the CESM CVCWG

CVDP-LE Data Repository, under https://www.cesm.ucar.edu/working_groups/CVC/cvdp-le/data-repository.html. The tigramite package for causal discovery is available under the public Github repository: https://github.com/jakobrunge/tigramite/. The code used to reproduce results and to plot most figures for this paper will be accessible at the time of publication of the manuscript in the following Github repository: https://github.com/EyringMLClimateGroup/karmouche22copernicus_CME_CMIP6.





## Appendix A

**Table A1.** Distribution of Pearson $r$ correlation values between the simulated (CMIP6 LE) and observed (ERA20C_ERA5, ERSSTv5) spatial patterns of PNA, PSA1, PDV, AMV and their Mean Score over the 1900-2014 period. Sorted by Alphabetical order.

| CMIP6 LE | Percentile | PNA (DJF) | PSA1 (ANN) | PDV (monthly) | AMV (monthly) | Mean Score |
|---|---|---|---|---|---|---|
| ACCESS-ESM1-5 | 10th | 0.80 | 0.57 | 0.68 | 0.67 | 0.67 |
| | 50th | 0.90 | 0.72 | 0.71 | 0.71 | 0.78 |
| | 90th | 0.93 | 0.79 | 0.77 | 0.75 | 0.80 |
| CESM2 | 10th | 0.84 | -0.64 | 0.82 | 0.68 | 0.61 |
| | 50th | 0.88 | 0.66 | 0.87 | 0.73 | 0.79 |
| | 90th | 0.91 | 0.77 | 0.88 | 0.77 | 0.82 |
| CNRM-ESM2-1 | 10th | 0.38 | 0.39 | -0.06 | 0.73 | 0.40 |
| | 50th | 0.59 | 0.53 | 0.68 | 0.74 | 0.64 |
| | 90th | 0.84 | 0.63 | 0.77 | 0.79 | 0.71 |
| CanESM5 | 10th | 0.76 | 0.56 | 0.75 | 0.68 | 0.73 |
| | 50th | 0.83 | 0.75 | 0.79 | 0.72 | 0.77 |
| | 90th | 0.88 | 0.81 | 0.82 | 0.76 | 0.80 |
| EC-Earth3 | 10th | 0.81 | -0.40 | 0.45 | 0.58 | 0.49 |
| | 50th | 0.85 | 0.69 | 0.71 | 0.63 | 0.73 |
| | 90th | 0.92 | 0.75 | 0.77 | 0.71 | 0.79 |
| GISS-E2-1-H | 10th | 0.73 | -0.70 | 0.73 | 0.62 | 0.46 |
| | 50th | 0.79 | -0.55 | 0.77 | 0.68 | 0.56 |
| | 90th | 0.86 | 0.73 | 0.81 | 0.72 | 0.76 |
| INM-CM5-0 | 10th | 0.53 | -0.08 | 0.47 | 0.62 | 0.48 |
| | 50th | 0.67 | 0.38 | 0.50 | 0.66 | 0.54 |
| | 90th | 0.73 | 0.57 | 0.56 | 0.71 | 0.60 |
| IPSL-CM6A-LR | 10th | 0.66 | 0.70 | 0.75 | 0.72 | 0.72 |
| | 50th | 0.73 | 0.80 | 0.79 | 0.76 | 0.77 |
| | 90th | 0.82 | 0.84 | 0.81 | 0.79 | 0.80 |
| MIROC6 | 10th | 0.81 | 0.68 | 0.83 | 0.67 | 0.77 |
| | 50th | 0.86 | 0.73 | 0.84 | 0.71 | 0.80 |
| | 90th | 0.91 | 0.78 | 0.85 | 0.74 | 0.82 |
| MPI-ESM1-2-LR | 10th | 0.76 | 0.70 | 0.75 | 0.64 | 0.74 |
| | 50th | 0.86 | 0.79 | 0.80 | 0.72 | 0.79 |
| | 90th | 0.93 | 0.82 | 0.83 | 0.77 | 0.82 |
| NorCPM1 | 10th | 0.38 | -0.58 | 0.72 | 0.64 | 0.41 |
| | 50th | 0.72 | -0.51 | 0.76 | 0.68 | 0.49 |
| | 90th | 0.82 | 0.58 | 0.79 | 0.72 | 0.70 |
| UKESM1-0-LL | 10th | 0.83 | 0.65 | 0.80 | 0.68 | 0.78 |
| | 50th | 0.88 | 0.75 | 0.82 | 0.74 | 0.80 |
| | 90th | 0.91 | 0.79 | 0.86 | 0.79 | 0.83 |



**Table A2.** Pearson correlation values obtained using 18 UKESM1-0-LL simulation, with respect to the observed (ERA20C_ERA5, ERSSTv5) spatial patterns of PNA, PSA1, PDV, AMV and their Mean Score over the 1900-2014 period. Sorted by mean score.

| UKESM1-0-LL Ensemble member | PNA (DJF) | PSA1 (ANN) | PDV (monthly) | AMV (monthly) | **Mean Score** |
|---|---|---|---|---|---|
| r19i1p1f2 | 0.91 | 0.84 | 0.86 | 0.80 | 0.86 |
| r6i1p1f3 | 0.89 | 0.75 | 0.86 | 0.78 | 0.83 |
| r3i1p1f2 | 0.90 | 0.76 | 0.85 | 0.76 | 0.83 |
| r14i1p1f2 | 0.94 | 0.75 | 0.82 | 0.68 | 0.82 |
| r2i1p1f2 | 0.88 | 0.78 | 0.84 | 0.75 | 0.82 |
| r1i1p1f2 | 0.87 | 0.81 | 0.82 | 0.78 | 0.82 |
| r11i1p1f2 | 0.91 | 0.68 | 0.81 | 0.76 | 0.81 |
| r8i1p1f2 | 0.90 | 0.78 | 0.79 | 0.70 | 0.80 |
| r17i1p1f2 | 0.80 | 0.76 | 0.84 | 0.81 | 0.80 |
| r7i1p1f3 | 0.86 | 0.78 | 0.82 | 0.72 | 0.80 |
| r4i1p1f2 | 0.90 | 0.67 | 0.82 | 0.73 | 0.80 |
| r16i1p1f2 | 0.86 | 0.75 | 0.81 | 0.72 | 0.79 |
| r10i1p1f2 | 0.88 | 0.74 | 0.83 | 0.66 | 0.79 |
| r9i1p1f2 | 0.89 | 0.56 | 0.82 | 0.77 | 0.79 |
| r18i1p1f2 | 0.88 | 0.60 | 0.82 | 0.76 | 0.78 |
| r5i1p1f3 | 0.85 | 0.75 | 0.81 | 0.68 | 0.78 |
| r13i1p1f2 | 0.84 | 0.71 | 0.80 | 0.74 | 0.78 |
| r12i1p1f2 | 0.79 | 0.67 | 0.80 | 0.68 | 0.74 |





**Table A3.** Cross-MCI and auto-MCI values calculated by PCMCI+ from reanalysis timeseries data for the complete 1900-2014 period. Values are relative to the complete period causal graph shown in Figure 2 (right panel) and Figure 5 (upper left panel). The table presents the cross-MCI (cross-correlation) values denoting the sign and strength of the causal link between node $i$ and node $j$ for lags between 0 and $\tau_{max}$. In bold are the highest absolute cross-MCI values for that specific link (detected within the statistical significance threshold, $\alpha_{pc} \leq 0.05$) and for which links are apparent on the causal graphs. The values are rounded to two decimal places.

| $i$ | $j$ | time lag $\tau$ 0 | 1 | 2 | 3 | 4 | 5 | 6 | 7 | 8 | 9 | 10 | 11 | 12 | 13 | 14 | 15 |
|---|---|---|---|---|---|---|---|---|---|---|---|---|---|---|---|---|---|
| **AMV** | AMV | 0.00 | 0.45 | 0.22 | 0.14 | 0.17 | 0.21 | 0.18 | 0.18 | 0.09 | 0.11 | 0.05 | 0.05 | 0.18 | 0.09 | 0.03 | 0.05 |
| | PNA | 0.18 | -0.20 | -0.06 | 0.01 | -0.13 | -0.11 | -0.17 | -0.04 | 0.01 | -0.04 | -0.13 | -0.18 | -0.17 | -0.19 | -0.16 | -0.06 |
| | PDV | -0.04 | -0.18 | -0.16 | -0.07 | -0.12 | -0.15 | -0.16 | -0.18 | -0.06 | -0.13 | -0.21 | **-0.25** | 0.02 | -0.18 | -0.06 | -0.11 |
| | PSA1 | **0.25** | -0.05 | -0.01 | -0.06 | -0.10 | -0.11 | -0.12 | -0.05 | -0.06 | -0.00 | -0.02 | -0.07 | -0.04 | -0.08 | -0.00 | 0.08 |
| **PNA** | AMV | 0.18 | -0.05 | -0.02 | -0.03 | -0.09 | -0.12 | -0.07 | 0.03 | -0.05 | 0.01 | 0.01 | -0.04 | 0.05 | 0.07 | 0.03 | 0.03 |
| | PNA | 0.00 | 0.06 | 0.05 | 0.04 | -0.04 | 0.10 | 0.02 | 0.00 | -0.06 | -0.11 | 0.03 | -0.14 | -0.01 | -0.06 | -0.11 | 0.05 |
| | PDV | **0.53** | -0.07 | 0.10 | -0.05 | -0.04 | 0.14 | 0.16 | -0.02 | -0.18 | -0.10 | -0.02 | 0.01 | 0.00 | -0.07 | -0.17 | -0.02 |
| | PSA1 | 0.11 | -0.09 | 0.06 | -0.12 | -0.08 | -0.01 | -0.03 | -0.05 | -0.13 | 0.00 | -0.13 | -0.11 | 0.12 | -0.12 | -0.10 | 0.13 |
| **PDV** | AMV | -0.04 | -0.03 | -0.01 | -0.04 | -0.09 | -0.15 | -0.09 | -0.05 | 0.00 | -0.02 | 0.02 | 0.02 | 0.01 | 0.10 | 0.12 | 0.14 |
| | PNA | **0.53** | 0.17 | 0.09 | 0.04 | 0.21 | 0.21 | 0.05 | -0.02 | -0.08 | -0.08 | 0.06 | 0.00 | -0.01 | -0.02 | -0.10 | -0.07 |
| | PDV | 0.00 | 0.33 | 0.18 | 0.07 | 0.18 | 0.18 | 0.08 | 0.02 | -0.12 | -0.01 | 0.09 | 0.11 | 0.02 | -0.09 | -0.21 | -0.15 |
| | PSA1 | -0.07 | 0.11 | 0.14 | 0.02 | 0.03 | 0.07 | 0.00 | -0.17 | -0.13 | -0.06 | -0.12 | -0.09 | -0.07 | -0.03 | -0.09 | 0.03 |
| **PSA1** | AMV | **0.25** | -0.09 | 0.15 | -0.09 | -0.07 | -0.07 | -0.06 | -0.21 | -0.16 | -0.14 | -0.19 | -0.17 | -0.16 | -0.20 | -0.19 | -0.15 |
| | PNA | 0.11 | -0.10 | 0.18 | -0.10 | 0.04 | 0.06 | 0.04 | 0.21 | 0.01 | 0.07 | -0.10 | 0.11 | 0.11 | -0.01 | -0.17 | -0.09 |
| | PDV | -0.07 | 0.03 | 0.11 | -0.01 | 0.02 | 0.09 | 0.09 | **0.23** | 0.08 | -0.07 | -0.13 | -0.07 | -0.08 | -0.10 | -0.21 | **-0.31** |
| | PSA1 | 0.00 | 0.04 | 0.18 | 0.03 | 0.19 | 0.04 | 0.10 | -0.02 | -0.16 | 0.02 | -0.09 | 0.01 | -0.09 | -0.19 | -0.19 | 0.01 |



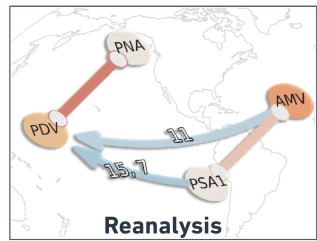

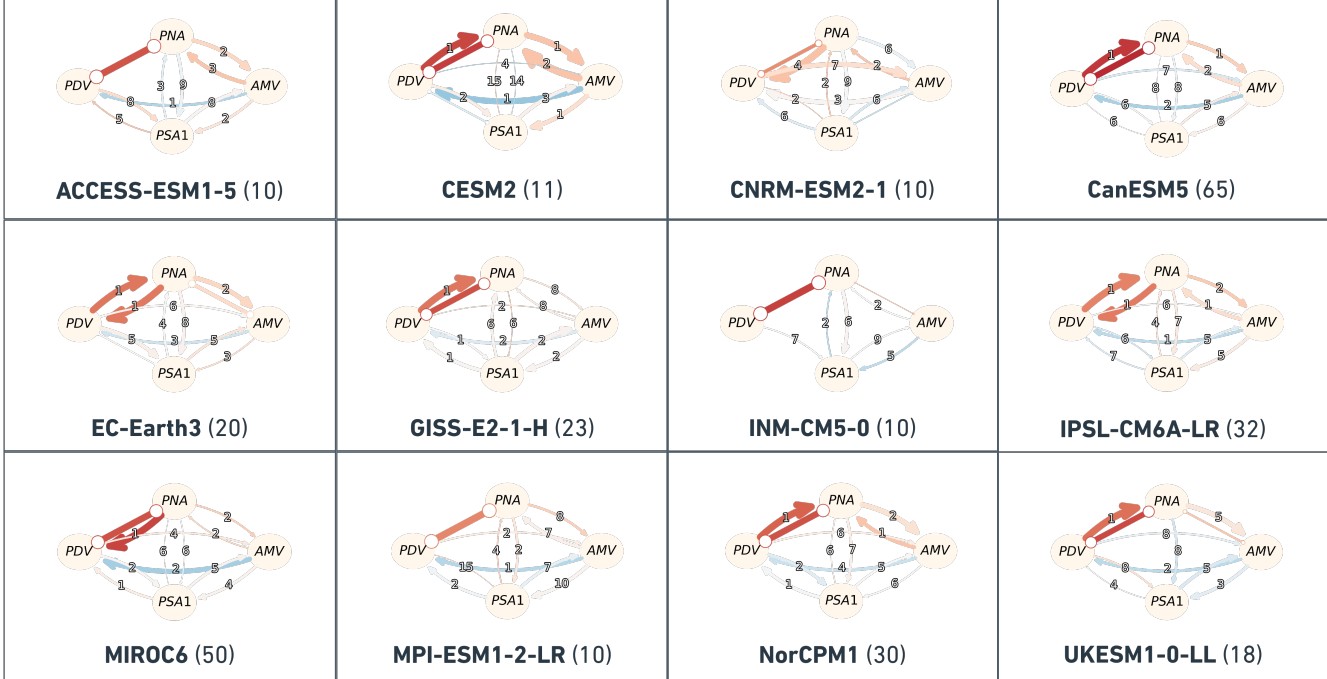

**Figure A1.** Similar to Figure 7 but for the 12 CMIP6 models during the **complete 1900-2014 period**. Each panel has a label stating the model name and the number of ensemble members between parenthesis. The auto-MCI values were not taken into consideration.



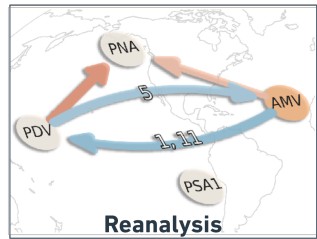

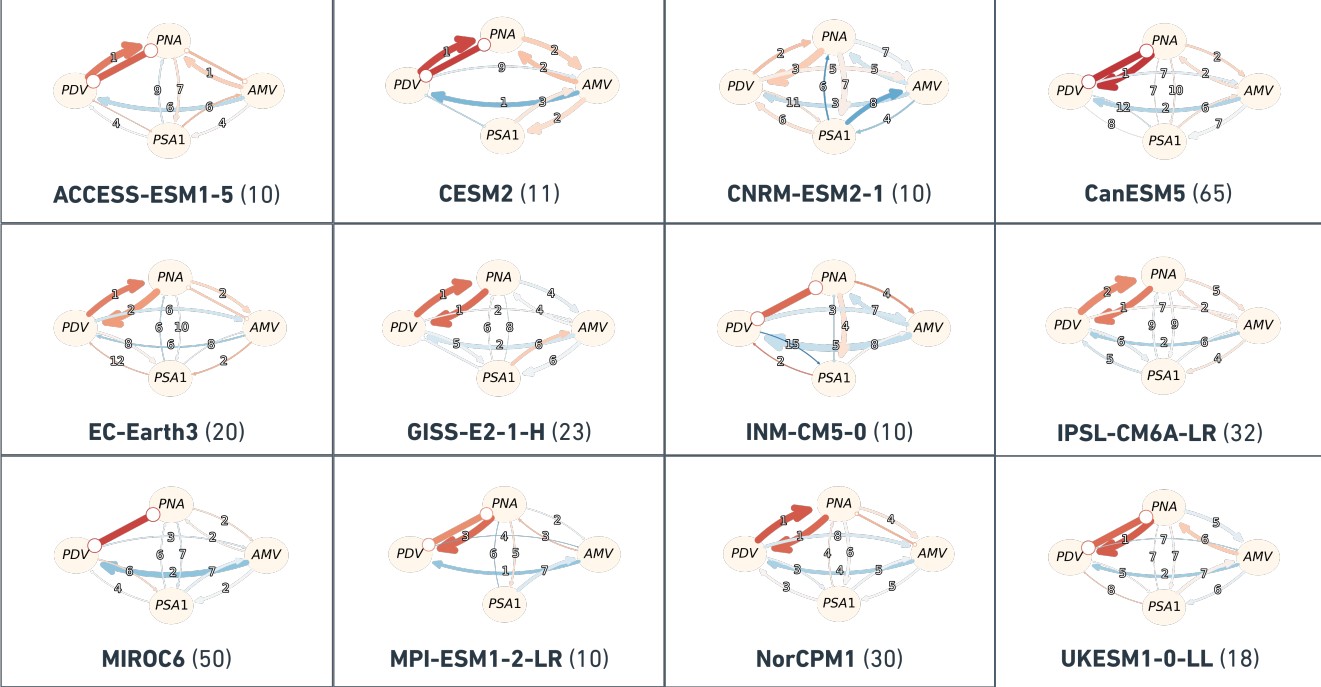

**Figure A2.** Similar to Figure 7 but for the 12 CMIP6 models during the **Out-of-Phase regime**. Each panel has a label stating the model name and the number of ensemble members between parenthesis. The auto-MCI values were not taken into consideration.



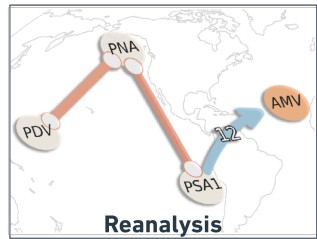

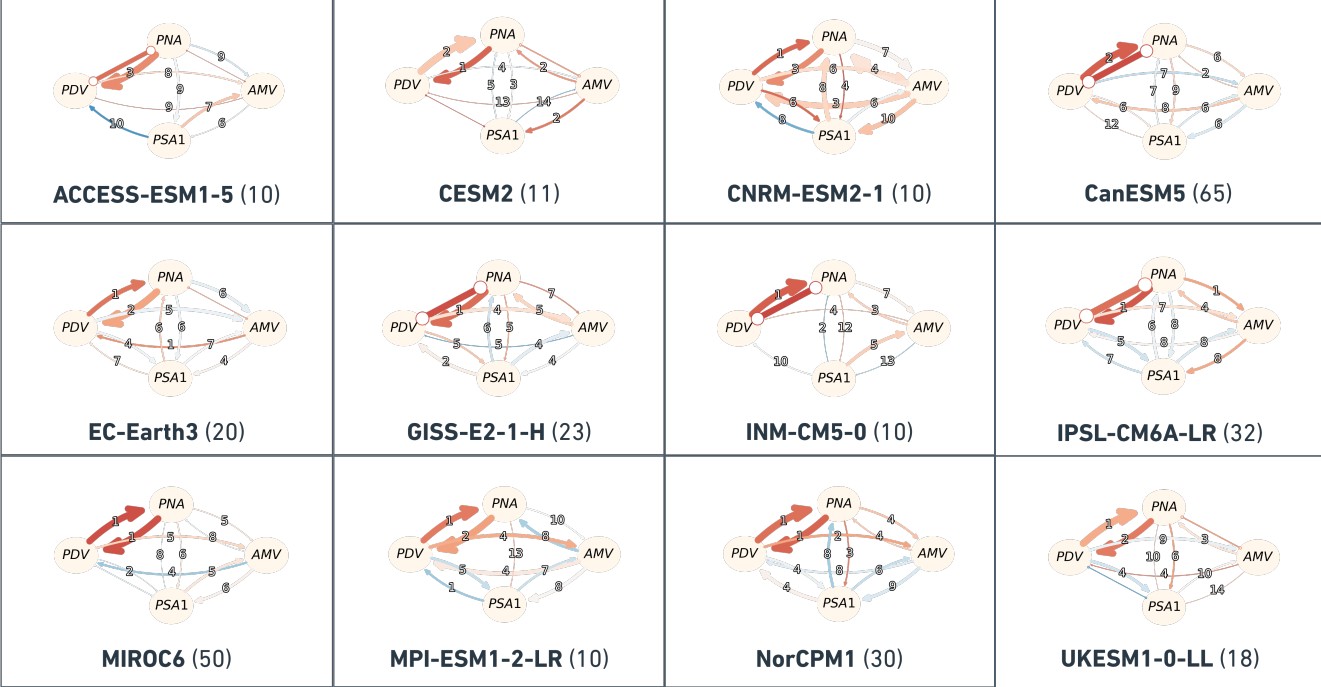

**Figure A3.** Similar to Figure 7 but for the 12 CMIP6 models during the **In-Phase regime**. Each panel has a label stating the model name and the number of ensemble members between parenthesis. The auto-MCI values were not taken into consideration.





*Author contributions.*  SK and VE designed and organized the study and lead the interpretation of the results. SK lead the writing of the manuscript, performed the data processing and analysis, and prepared all figures and tables. EG contributed to the concept of the study and supported the analysis. JR developed a new version of the causal discovery tool that supported this study. JM, AP, and KW contributed to the interpretation of the results. All coauthors contributed to the writing of the manuscript.

*Competing interests.*  The authors declare that they have no conflict of interest.

*Acknowledgements.*  Funding for this study was provided by the European Research Council (ERC) Synergy Grant "Understanding and modeling the Earth System with Machine Learning (USMILE)" under the Horizon 2020 research and innovation program (Grant agreement No. 855187) and the "Advanced Earth System Model Evaluation for CMIP (EVal4CMIP)" project funded by the Helmholtz Society. We acknowledge the World Climate Research Program's (WCRP's) Working Group on Coupled Modelling (WGCM), which is responsible for CMIP, and we thank the climate modelling groups listed in Table 1 for producing and making available their model output. This work used
resources of the Deutsches Klimarechenzentrum (DKRZ) granted by its Scientific Steering Committee (WLA) under project ID bd1083.



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
