# Peer review of "Regime-oriented causal model evaluation of Atlantic-Pacific teleconnections in CMIP6"

_EGUsphere, 2022_

## Referee Comment (RC2)

Review of "Regime-oriented causal model evaluation of Atlantic-Pacific teleconnections in CMIP6" by Karmouche et al.:

The authors utilize the PCMCI causality approach to investigate the interanual Atlantic-Pacific teleconnections, then they compare the causal relations in observations with that in CMIP6 model simulations. I think that there are two advantages of this work: (1) Analyzing interanual Atlantic-Pacific teleconnections is an important task in climate science, and it is novel to include the state-of-art PCMCI causality method. Additionally, when interpreting the causal detection results, the authors cite and combine enough previous studies to explain the mechanisms within Atlantic-Pacific teleconnections. (2) The authors present an example for using the PCMCI to evaluate the CMIP6 model simulations, and they demonstrate that this way can help to extract more useful information than the conventional spatial-pattern-based evaluation method.

Overall, this work is scientific and worth to be published. But it is necessary for the authors to carefully revise the manuscript before it is accepted for publication on Earth System Dynamics. Please find my comments and suggestions below.

1.  The Atlantic-Pacific teleconnections are dependent on the specific time scales, e.g., the MJO-NAO teleconnection on intraseasonal time scale, and the PDV-AMV teleconnection on interanual or decadal time scale. The current manuscript may only focus on the interanual time scale, such that the authors may consider to confine the title of the article into interanual Atlantic-Pacific teleconnections.

2.  As the authors mention, the length of the time series can influence the performance of causal detections. The authors should provide a table to list the used length of time series for every cases which they analyzed in the manuscript, including the observation and model data under different cases. Especially, when analyzing the same causal link such as the PDV-AMV link under out-of-phase regime, please put the used time series lengths of observation and model data together, which is benefit for comparisons. This supplementary is necessary because the readers can use this information to judge how convincing the results are, and whether their data lengths are enough if they would also utilize the PCMICI to analyze their own work.

3.  The authors focuses on the regime-dependent causal interactions. This is a very good attempt because the nonlinearity nature of the climate system determines that the climatic interactions are varied with the different temporal regimes. However, I am a few concerned whether their technical processing on the time series will influence the accuracy of causal inference. My concern is as following.

As previous studies suggested (Smirnov and Bezruchko, 2012; Smirnov, 2013), the low temporal resolution or resampling of the time series can lead to spurious causalities within the underlying variables. The authors divide the PDV/AMV/PNA/PSA time series into several segments within different regimes, then combine the time series segments within in/out-of-phase regimes. This is actually a resampling processing on the time series, thus is it possible to lead to spurious causalities as Smirnov and Bezruchko (2012) found? I agree that this issue is very challenging in techniques, but at least this should be mentioned and discussed in the manuscript. Otherwise the readers who are not expect in causal detection may neglect this issue, and this is not conductive to the method development of data-driven causal detections in climate science.

Reference

[1] Smirnov D., Bezruchko B. 2012. Spurious causalities due to low temporal resolution: Towards detection of bidirectional coupling from time series. EPL, 100, 10005. https://iopscience.iop.org/article/10.1209/0295-5075/100/10005

[2] Smirnov D. 2013. Spurious causalities with transfer entropy. Phys. Rev. E, 87, 042917. https://journals.aps.org/pre/abstract/10.1103/PhysRevE.87.042917

4.  In line 252, it read as "This is to quantify the similarity between the observed and simulated spatial patterns for the four modes and build credibility that the CMIP6 simulated indices we use in the regime-oriented model evaluation have spatial expressions that resemble those of indices calculated from reanalysis datasets." This sentence is too long such that I cannot understand its meaning. Please rephrase it.

5.  In line 352, it reads as "The limitation presented by the length of unmasked time series during specific short regimes is eliminated when combining them." I would like to know more technical details about "combining them". Do you glue the time series segments during specific regimes and then calculate the lagged correlations and partial correlations? Please provide a little more explanations of your processing here.

6.  In figure 6, there is no unit for the horizontal axis, is it [years]?

7.  In line 494, it reads as "The model doesn't only have the lowest level ...". "doesn't" should be corrected as "does not".

---

## Author Response (AR1)

**Author Response**

**Reply to Christian Franzke (Editor):**

We are grateful to the editor for the guidance throughout the submission process and finding two experienced reviewers. Their reviews and comments have been constructive, leading us to introduce changes to the manuscript based on their feedback. We have furnished a point-by-point response to each review and comment in our response document. The revised version of the manuscript also addresses notifications from Polina Shvedko regarding the copyrights statement, the caption style and the coloured table.

**Reply to Referee Comment #1:**

We thank the referee for his/her constructive comments. We copy the Referees' Comments between "quotation marks" and we address them below in **bold**. We use the following notation: P1 L1 means Page 1, Line 1. We use blue colour for the text to be added to the new version of the manuscript. Regarding the referee comment, we understand that the comment contains two main points that we would like to address separately

Answer to: **"**In lines 36-37 it is mentioned that causal graphs help to determine if a specific phenomenon is simulated for the right reasons. In that sense, it would be useful to see more discussion on the reasons behind the degree of similarity with the observed AMV and PDV connections. In particular, some discussion about the physical reasons. This would demonstrate the correct use of any statistical method which should always be justified by the physics behind the phenomena under analysis**.** […]**"**

**We actually believe that the discussion of the physical reasons behind the AMV and PDV is fairly documented throughout the paper. We discuss the physical processes mainly for the results from reanalysis datasets which we believe capture the natural mechanisms the best. The scope of the paper is to present causal discovery as a tool to evaluate the causal dynamics (laid down by physical processes) within climate models but with respect to reference causal dynamics from analysis of reanalysis data. Therefore, in Introduction, in P3 L62-76, we cite a number of previous studies on the Atlantic-Pacific interactions which investigated the AMV-PDV link. We also introduce the possible extra-tropical teleconnections that might play a role in these interactions (PNA and PSA). In Results Section 3.2, we show and analyse the causal networks constructed from reanalysis datasets and in P17-18 L361-399 we discuss the estimated AMV and PDV links with respect to previous findings on the physical mechanisms behind those links. However, we further address the comment by including the following (in blue) in the Discussion section in P26 L517:**

**[…] LEs display the highest degree of similarity.** **During the analysis of the complete 1900-2014 period, a considerable fraction of simulations belonging to these CMIP6 models estimates an opposite-sign response from AMV to PDV (represented by blue AMV→PDV links, see Figure A1). The clear occurrence of this opposite-sign response in several CMIP6 LEs (notably, CESM2, CanESM5, MIROC6 and MPI-ESM1-2-LR) shows that these models realistically simulate the mechanisms that connect Atlantic and Pacific modes of SST variability. The direct connection between the**

**Atlantic and Pacific basins involves mainly the tropical Walker circulation and its associated SST, evaporation, wind and SLP changes where rising temperatures in the Atlantic Ocean can cause a cooling effect similar to La Niña in the equatorial Pacific (McGregor et al., 2014; Kucharski et al., 2016; Li et al., 2016; Ruprich-Robert et al., 2021; Meehl et al., 2021a). Moreover, these […]**

**With the following addition to the bibliography:**

Ruprich-Robert, Y., Moreno-Chamarro, E., Levine, X., Bellucci, A., Cassou, C., Castruccio, F., Davini, P., Eade, R., Gastineau, G., Hermanson, L., et al.: Impacts of Atlantic multidecadal variability on the tropical Pacific: a multi-model study, npj climate and atmospheric science, 4, 1–11, 2021

Answer to: **"**[…] In this article as well as Meehl et al. (2021) it is pointed out that the short observational record is a limit to study the AMV PDV connections. Therefore, the use of climate models is essential to draw conclusions on the nature of these connections. While this paper focuses on the identification of these connection on large ensembles, more discussion is needed about the relevant biases that limit the correct representation of this connection in climate models, particularly because there are studies which report that most models present strong biases in representing the Pacific-Atlantic interbasin interactions (e.g. Kajtar et al. 2018; McGregor et al. 2018).**"**

**We find this comment important to address, especially with regards to the persisting biases over the tropical Atlantic that certainly reflect the ability of models to simulate the observed AMV. The results of pattern correlations showed that most CMIP6 models simulate better the PDV patterns compared to AMV patterns with respect to reanalysis. We further address this issue in the Discussion section in P28 L553:**

"However, despite the improvement of CMIP6 models in capturing the different modes of climate variability (Fasullo et al., 2020) recent studies already pointed to persisting tropical Atlantic biases that knew little or no improvement compared to CMIP5 (Richter and Tokinaga, 2020; Farneti et al., 2022). These biases certainly affect the simulation of Atlantic variability within CMIP6 models as they project additional uncertainties on the AMV-related causal dynamics and spatial patterns. Moreover, previous research showed that, on the decadal timescale, Atlantic mean SST biases in CMIP5 models are directly related to the variability of trade winds over the region (Kajtar et al. 2018). McGregor et al. (2018) showed that the addition of the CMIP5 Atlantic bias leads to enhanced descending motion trends in the western and eastern Pacific, and reduced trend in the central Pacific. The same study found that the observed northward

migration of the Intertropical Convergence Zone (ITCZ) is absent when introducing CMIP5 Atlantic bias."

The references in blue below have also been added to the attached revised manuscript.

Farneti, R., Stiz, A., and Ssebandeke, J. B.: Improvements and persistent biases in the southeast tropical Atlantic in CMIP models, npj Climate and Atmospheric Science, 5, 1–11, 2022.

Fasullo, J. T., Phillips, A. S., and Deser, C.: Evaluation of leading modes of climate variability in the CMIP archives, Journal of Climate, 33, 5527–5545, 2020

Kajtar, J. B., A. Santoso, S. McGregor, M. H. England, and Z. Baillie, 2018: Model under-representation of decadal Pacific trade wind trends and its link to tropical Atlantic bias. Clim. Dyn., 50, 1471–1484, https://doi.org/10.1007/s00382-017-3699-5.

McGregor, S., Stuecker, M. F., Kajtar, J. B., England, M. H., and Collins, M.: Model tropical Atlantic biases underpin diminished Pacific decadal variability, Nature Climate Change, 8, 493–498, 2018

Richter, I. and Tokinaga, H.: An overview of the performance of CMIP6 models in the tropical Atlantic: mean state, variability, and remote impacts, Climate Dynamics, 55, 2579–2601, 2020

**Reply to Referee Comment #2:**

We thank the referee for his/her constructive comment. We copy the Referees' Comments between "quotation marks" and we address them below in bold. We use the following notation: P1 L1 means Page 1, Line 1. We use blue colour for the text to be added to the new version of the manuscript. We understand that the comment consists of seven main points and we would like to address them separately as follows:

Answer to: "1. The Atlantic-Pacific teleconnections are dependent on the specific time scales, e.g., the MJO-NAO teleconnection on intraseasonal time scale, and the PDV-AMV teleconnection on interanual or decadal time scale. The current manuscript may only focus on the interanual time scale, such that the authors may consider to confine the title of the article into interanual Atlantic-Pacific teleconnections."

**In our study the causal connections among PDV, AMV and their extra-tropical teleconnections obtained from reanalyses data and CMIP6 simulations are detected with lags ranging up to 15 years. This implies that the detected connections go beyond the interannual time scale and reach the decadal time scale (for example see Fig. 5). Thus, we believe that confining the title to "interannual" might be misleading to the reader.**

Answer to: "2. As the authors mention, the length of the time series can influence the performance of causal detections. The authors should provide a table to list the used length of time series for every cases which they analyzed in the manuscript, including the observation and model data under different cases. Especially, when analyzing the same causal link such as the PDV-AMV link under out-of-phase regime, please put the used time series lengths of observation and model data together, which is benefit for comparisons. This supplementary is necessary because the readers can use this information to judge how convincing the results are, and whether their data lengths are enough if they would also utilize the PCMICI to analyze their own work."

**We agree that this can be of great importance to the reader and we included in Appendix Tables (A4-A10) with the length of regimes for every model and simulation we use. To make comparison easier, we kept the regime lengths for reanalysis data on top of each table. We also add the following sentence to refer to these additional Appendix Tables in P9 L209:**

**"We show in Appendix Table A4 to Table A10, the number of years per regime for each dataset analysed."**

Answer to "3. The authors focuses on the regime-dependent causal interactions. This is a very good attempt because the nonlinearity nature of the climate system determines that the climatic interactions are varied with the different temporal regimes. However, I am a few concerned whether their technical processing on the time series will influence the accuracy of causal inference. My concern is as following. As previous studies suggested (Smirnov and Bezruchko, 2012; Smirnov, 2013), the low temporal resolution or resampling of the time

series can lead to spurious causalities within the underlying variables. The authors divide the PDV/AMV/PNA/PSA time series into several segments within different regimes, then combine the time series segments within in/out-of-phase regimes. This is actually a resampling processing on the time series, thus is it possible to lead to spurious causalities as Smirnov and Bezruchko (2012) found? I agree that this issue is very challenging in techniques, but at least this should be mentioned and discussed in the manuscript. Otherwise the readers who are not expect in causal detection may neglect this issue, and this is not conductive to the method development of data-driven causal detections in climate science."

> **As explained in Section 2.1.3, the regimes are simply masks that are applied on time series arrays to filter out periods of interest based on the sign of the low-pass filtered AMV and PDV (see Figure 3). On the resampling issue, we acknowledge that the sample size is very important for every statistical analysis and that the yearly averaging we perform for AMV and PDV indices might lead to spurious links. Thus, we included the following for more clarification in P9 L209 right after the addition related to previous point of the comment:**

> **"Nonetheless, it should be stated that the results of the regime-oriented causal analysis account for potential errors related to the sampling of the data. A study from Smirnov and Bezruchko (2012) demonstrated, using a variety of examples, how sampling at lower intervals can produce large "spurious" results."**

> **With the following addition to the bibliography:**

> **Smirnov, D. and Bezruchko, B.: Spurious causalities due to low temporal resolution: Towards detection of bidirectional coupling from time series, EPL (Europhysics Letters), 100, 10 005, 2012.**

Answer to: "4. In line 252, it read as "This is to quantify the similarity between the observed and simulated spatial patterns for the four modes and build credibility that the CMIP6 simulated indices we use in the regime-oriented model evaluation have spatial expressions that resemble those of indices calculated from reanalysis datasets." This sentence is too long such that I cannot understand its meaning. Please rephrase it"

> **This is now split into two sentences as follows:**

> **"This is to quantify the similarity between the observed and simulated spatial patterns for each of the four modes of climate variability we are to analyse.  The purpose is to check if the CMIP6-simulated indices have spatial expressions that resemble those of indices calculated from reanalysis datasets."**

Answer to: "5. In line 352, it reads as "The limitation presented by the length of unmasked time series during specific short regimes is eliminated when combining them." I would like to know more technical details about "combining them". Do you glue the time series

segments during specific regimes and then calculate the lagged correlations and partial correlations? Please provide a little more explanations of your processing here."

**To better clarify we adjusted text in P17 L352-353 into:**

**"The limitation presented by the**  **fact that some regimes might be too short to detect any causal links (e.g. PDV-/AMV-, 25 years) is overcome when introducing causal graphs for In-Phase and Out-of-Phase regimes (panels in the bottom of Figure 5** **). As explained in Sect. 2.1.3, the In-Phase regime is made up of the time-steps where AMV and PDV happen to be on the same phase (PDV+/AMV+ and PDV-/AMV-) while the Out-of-Phase regime is composed of time-steps where the two modes are on opposite phases (PDV+/AMV- and PDV-/AMV+), resulting in longer regime periods."**

Answer to: "6. In figure 6, there is no unit for the horizontal axis, is it [years]?"

**Yes, the unit is in [years] and we added it to Figure 6. The figure caption is to be slightly adjusted to now read:**

**"Lagged correlations of original data (lag in years, complete period graph from reanalysis data) in red shown together with lagged correlations of an ensemble of synthetic data generated by a linear Gaussian structural causal model with causal coefficients and noise structure estimated from the original data. The mean lagged correlations from the synthetically generated data are shown in black, and their 5th-95th percentile range in grey."**

Answer to: "7. In line 494, it reads as "The model doesn't only have the lowest level ...". "doesn't" should be corrected as "does not"."

**This has been fixed as suggested and now reads: "The model** **does not only have ..."**